# Global analysis of gene expression reveals mRNA superinduction is required for the inducible immune response to a bacterial pathogen

Kevin C Barry[1], Nicholas T Ingolia[2]*, Russell E Vance[1,3,4]*

[1]Division of Immunology and Pathogenesis, Department of Molecular and Cell Biology, University of California, Berkeley, Berkeley, United States; [2]Division of Biochemistry, Biophysics and Structural Biology, Department of Molecular and Cell Biology, University of California, Berkeley, Berkeley, United States; [3]Cancer Research Laboratory, University of California, Berkeley, Berkeley, United States; [4]Howard Hughes Medical Institute, University of California, Berkeley, Berkeley, United States

*For correspondence: ingolia@berkeley.edu (NTI); rvance@berkeley.edu (REV)

**Competing interests:** The authors declare that no competing interests exist.

**Abstract** The inducible innate immune response to infection requires a concerted process of gene expression that is regulated at multiple levels. Most global analyses of the innate immune response have focused on transcription induced by defined immunostimulatory ligands, such as lipopolysaccharide. However, the response to pathogens involves additional complexity, as pathogens interfere with virtually every step of gene expression. How cells respond to pathogen-mediated disruption of gene expression to nevertheless initiate protective responses remains unclear. We previously discovered that a pathogen-mediated blockade of host protein synthesis provokes the production of specific pro-inflammatory cytokines. It remains unclear how these cytokines are produced despite the global pathogen-induced block of translation. We addressed this question by using parallel RNAseq and ribosome profiling to characterize the response of macrophages to infection with the intracellular bacterial pathogen *Legionella pneumophila*. Our results reveal that mRNA superinduction is required for the inducible immune response to a bacterial pathogen.

## Introduction

Gene expression is a concerted process that is regulated at multiple steps, including transcription, mRNA degradation, translation, and protein degradation. Most global studies of gene expression have focused on the transcriptional response, but the relative importance of transcription in determining protein levels remains debated (*Li et al., 2014*; *Schwanhäusser et al., 2011*; *Breker and Schuldiner, 2014*; *Maier et al., 2009*; *Vogel and Marcotte, 2012*; *de Sousa Abreu et al., 2009*). One recent study analyzed the response of dendritic cells to lipopolysaccharide (LPS) and found that changes in mRNA levels accounted for ~90% of observed alterations in protein levels (*Jovanovic et al., 2015*). However, the response to infection with a virulent pathogen is certainly more complicated than the response to a purified immunostimulatory ligand such as LPS. Indeed, pathogens have evolved to disrupt or manipulate almost every cellular process involved in gene expression (*Finlay and McFadden, 2006*). An effective innate immune response to infection therefore requires that host cells be able to induce appropriate responses in the face of pathogen manipulation.

**eLife digest** We are constantly exposed to microbes that are capable of causing disease, but our immune system is generally able to protect us by producing specific proteins that help kill the microbes. In response, many infectious microbes have developed ways to obstruct the immune system of their host. For example, a bacterium called *Legionella pneumophila* – which can cause serious lung infections – blocks the ability of host immune cells to generate new proteins.

To make a new protein, genetic information in the form of DNA is first copied to make molecules called messenger ribonucleic acids (or mRNAs for short). These molecules are then used as templates to make the protein. Despite the fact that *L. pneumophila* is capable of interfering with this vital process, the host is still able to mount a protective immune response. It was not clear how this is possible.

To address this question, Barry et al. studied immune cells from mice that had been infected with *L. pneumophila*. The experiments show that these immune cells produce large amounts of mRNAs that correspond to proteins needed for the immune response. These mRNAs overwhelm the protein production block imposed by the bacteria, allowing the immune cells to produce these proteins and trigger an immune response.

The experiments suggest that, in response to microbes that block the production of proteins, changes in the amount of mRNA in a cell are the strongest indicators of how much protein the cells will be able to produce. These findings shed new light onto how the immune system can overcome interference by microbes to protect the host.

Inhibition of host protein synthesis is a common strategy used by many viral and bacterial pathogens to disrupt host gene expression (*Mohr and Sonenberg, 2012*; *Lemaitre and Girardin, 2013*). For example, the intracellular bacterial pathogen *L. pneumophila* uses its Dot/Icm type IV secretion system (T4SS) to translocate into host cells several effector proteins that block host protein synthesis, including at least four effectors that target the elongation factor eEF1A (*Lemaitre and Girardin, 2013*; *Barry et al., 2013*; *Belyi et al., 2008*; *Fontana et al., 2011*; *Shen et al., 2009*). Similarly, the bacterial pathogen *Pseudomonas aeruginosa* blocks host translation elongation by secretion of exotoxin A (*Lemaitre and Girardin, 2013*; *Dunbar et al., 2012*; *Iglewski et al., 1977*). Interestingly, we previously discovered that host cells respond to protein synthesis inhibition — whether by *Legionella*, exotoxin A, or by pharmacological agents that block translation initiation or elongation — by initiating a specific host response characterized by production of specific pro-inflammatory cytokines, including interleukin-23 (*Il23a*), granulocyte macrophage colony-stimulating factor (*Csf2*) and interleukin-1α (*Il1a*) (*Barry et al., 2013*; *Fontana et al., 2011*). The mechanism by which infected host cells are able to produce certain cytokines despite a global (>90%) block in protein synthesis remains unclear, but at least two distinct models have been proposed (*Mohr and Sonenberg, 2012*; *Lemaitre and Girardin, 2013*; *Barry et al., 2013*; *Fontana et al., 2011*; *Dunbar et al., 2012*; *Fontana and Vance, 2011*; *McEwan et al., 2012*; *Chakrabarti et al., 2012*). One model posits that the block in protein synthesis leads to superinduction of cytokine mRNAs that is sufficient to overcome the partial block in host protein synthesis (*Barry et al., 2013*; *Fontana et al., 2011*). Alternatively, it has been proposed that host cells may circumvent the global block in protein synthesis by selective translation of specific cytokine transcripts (*Dunbar et al., 2012*; *Asrat et al., 2014*).

To determine how host cells mount an inflammatory response when protein synthesis is disrupted, we performed parallel RNAseq and ribosome profiling (*Ingolia et al., 2012*, *2009*, *2011*) of *Legionella*-infected mouse primary bone-marrow-derived macrophages (BMMs). The results reveal the relative contributions of translational regulation and mRNA induction in controlling immune responses to pathogenic *L. pneumophila*, and support a model in which the majority of gene induction in response to pathogenic infections occurs at the level of mRNA induction. We were able to identify a subset of mRNAs that display higher-than-average ribosome occupancy, but the elevated occupancy of these mRNAs was observed in uninfected cells as well as in cells infected with *L. pneumophila*. We propose that mRNA superinduction provides a robust mechanism for host cells to initiate a response to infection despite pathogen-mediated disruption of host gene expression.

## Results

### mRNA superinduction mediates the host response to virulent *L. pneumophila*

The relative role of transcription versus translation in mediating the inducible response to an infection with a virulent bacterial pathogen remains unclear. Thus, we performed ribosome profiling (*Ingolia et al., 2012*, *2009*, *2011*) and total (rRNA-depleted) RNA sequencing of BMMs infected with *L. pneumophila*. BMMs were infected with a virulent *ΔflaA* strain, an avirulent T4SS-deficient *ΔdotAΔflaA* strain, or a *Δ7ΔflaA* strain that lacks the seven effectors associated with inhibition of host protein synthesis. RNA was isolated at 6 hr post-infection, which was the earliest we could detect significant *L. pneumophila*-induced translation inhibition without marked cytotoxicity (data not shown). *L. pneumophila* strains on the *ΔflaA* background were used to reduce cell cytotoxicity by avoiding the effects of NAIP5/NLRC4 inflammasome activation by flagellin (*Molofsky et al., 2006*; *Ren et al., 2006*) and we previously showed loss of flagellin does not affect blockade of host translation or the transcriptional induction of inflammatory cytokines (*Barry et al., 2013*). Control experiments demonstrated that ~90% of macrophages were infected with at least one bacterium under our infection conditions (*Figure 1—figure supplement 1A–B*).

Lysates from infected macrophages were split and used to generate ribosome profiling libraries and RNAseq libraries, thereby allowing us to compare directly the mRNA levels and ribosome occupancy of those mRNAs from the same cells. As a confirmation of the quality of the ribosome profiling libraries, ribosome footprints were found to map preferentially to the exonic regions of infection-induced genes (*Figure 1*), and showed a strong bias toward 27–28 nucleotide fragment lengths (*Figure 1—figure supplement 2*), consistent with the known size of ribosome-protected footprints. In accord with previous studies, induction of ribosome footprints on *Gem*, *Csf2*, and *Il23a* required the seven-bacterial effectors associated with the block in host protein synthesis, while induction of ribosome footprints corresponding to *Il1a* and *Il1b* required the bacterial T4SS (*Figure 1A–F*).

We first analyzed WT BMMs for T4SS-dependent gene induction, defined as the ratio of normalized read counts in the virulent *ΔflaA* to avirulent *ΔdotAΔflaA L. pneumophila*-infected conditions (see Materials and methods). We found that the majority of T4SS-dependent increases in ribosome footprints could be explained at the level of mRNA induction, as there was nearly a perfect linear correlation between the extent of mRNA induction and ribosome footprints for all T4SS-induced genes (*Figure 2A*). This correlation held for numerous known pathogen-induced mRNAs, including *Il23a*, *Gem*, *Csf2*, *Il6*, *Tnf*, *Cxcl1*, *Cxcl2*, *Dusp1*, and *Dusp2*, as well as for the cytokines *Il1a* and *Il1b* that were previously proposed to be preferentially translated (*Asrat et al., 2014*). To confirm that cytokine protein levels correlate with mRNA levels, we infected BMMs with *ΔflaA* or *ΔdotAΔflaA L. pneumophila* and measured the levels of 42 cytokines or immune-related proteins in the supernatants or cell lysates of these BMMs at 6 hr, using commercially available bead arrays. Of the cytokines assayed, 18 cytokines/proteins were measured above the limit of detection in lysates, and 22 cytokines/proteins were measured above the limit of detection in supernatants. The T4SS-dependent fold-induction of these protein levels was plotted versus the T4SS-dependent fold-induction of mRNA levels (*Figure 2B–C*). We observed a robust correlation between the extent of mRNA induction and the extent of protein induction, particularly in lysates (*Figure 2B*). The correlation seems to apply for the most highly induced proteins/mRNAs (e.g. IL-10 (*Il10*) and GM-CSF (*Csf2*)) but also for more modestly induced cytokines (*Il1a*, *Il1b*, *Cxcl10*). The less robust correlation between mRNA levels and protein levels in the cell supernatant (*Figure 2C*) may reflect differing rates of secretion, accumulation in the supernatant over time, re-binding to cell surface receptors, and stability in the supernatant. Taken together, these results suggest that the inducible immune response to *L. pneumophila* is controlled primarily at the level of mRNA superinduction (*Figure 2*).

### mRNA induction accounts for effector-induced gene expression

*L. pneumophila* uses multiple mechanisms to block host protein synthesis. It has been shown that up to seven bacterial effectors secreted into the host cytosol can block translation (*Barry et al., 2013*; *Fontana et al., 2011*). Interestingly, the *Δ7* strain that lacks these effectors is still able to partially suppress host protein synthesis by a mechanism that remains to be fully characterized (*Barry et al.,*

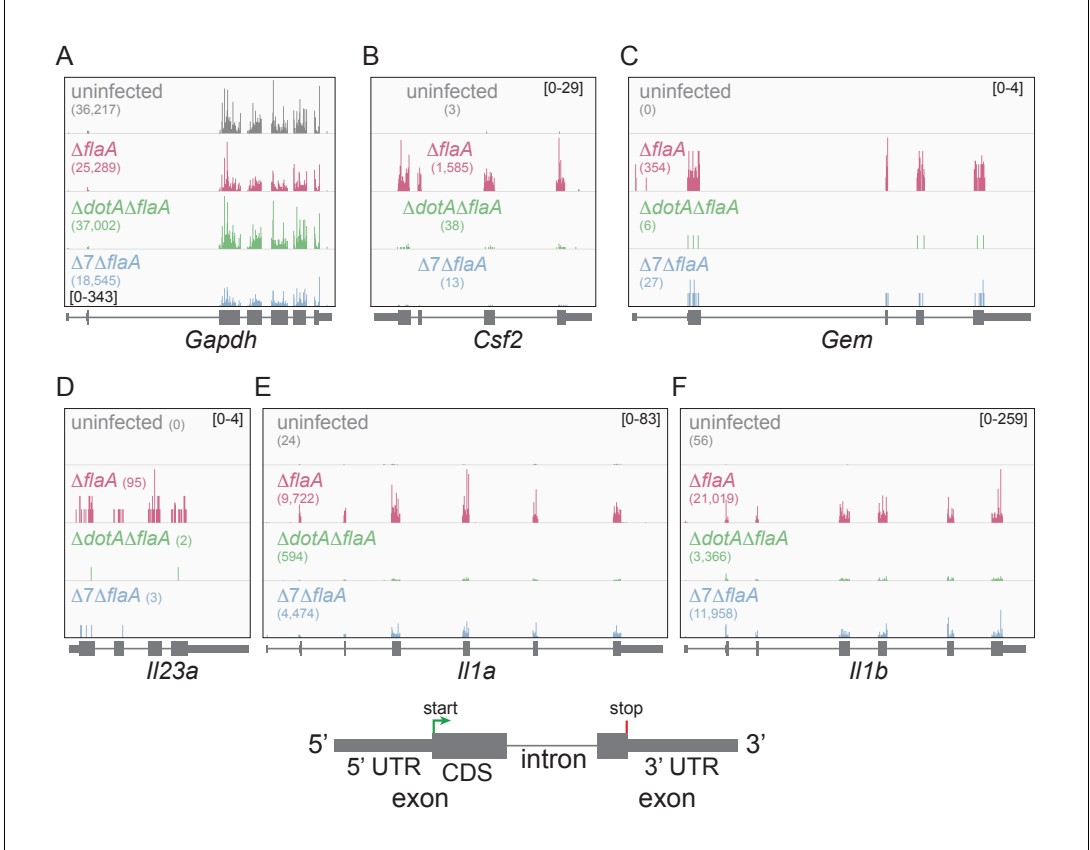

**Figure 1.** Mapping of ribosome profiling reads to the genomic sequence of specific *L. pneumophila*-induced genes of interest. (**A–F**) Ribosome footprint reads were mapped to the genome and the number of footprints on the mRNAs for *Gapdh* (A), *Csf2* (B), *Gem* (C), *Il23a* (D), *Il1a* (E), and *Il1b* (F) was visualized. Numbers in parentheses show the total read count of ribosome footprints found on the indicated transcript. Bracketed numbers represent read count data range. Gray, uninfected BMMs. Red, *ΔflaA*-infected BMMs. Green, *ΔdotAΔflaA*-infected BMMs. Blue, *Δ7ΔflaA*-infected BMMs.

The following figure supplements are available for figure 1:

**Figure supplement 1.** Quanification of *L. pneumophila* infectivity.

**Figure supplement 2.** Ribosome profiling libraries show a strong bias in size distribution.

*2013*; *Fontana et al., 2011*; *Ivanov and Roy, 2013*). It has been proposed that T4SS-competent *L. pneumophila* damages host cell membranes, resulting in ubiquitylation-dependent downregulation of mTOR activity and a block in cap-dependent translation (*Ivanov and Roy, 2013*). Consistent with its ability to partially suppress protein synthesis, the *Δ7* strain still provokes IL-1α production, although its ability to stimulate *Il23a* and *Csf2* expression is diminished (*Barry et al., 2013*; *Fontana et al., 2011*).

To determine the mechanism of effector-triggered cytokine induction, we performed parallel RNAseq and ribosome profiling of BMMs infected either with *ΔflaA* or *Δ7ΔflaA L. pneumophila*. As expected, induction of *Gem*, *Il23a*, and *Csf2* is highly dependent on the seven bacterial effectors, but again, similar to the total T4SS-dependent gene induction (*Figure 2*), the seven effector-dependent induction of ribosome footprints on these genes could be explained at the level of mRNA induction (*Figure 3A*). While the seven effector-dependent induction of the genes *Dusp1*, *Dusp2*, *Cxcl1*, *Cxcl2*, *Tnf*, *Il1a*, *Il1b*, and *Il6* was low, all changes in ribosome footprint reads could again be explained by changes in mRNA levels (*Figure 3A*). These data suggest that T4SS-dependent and

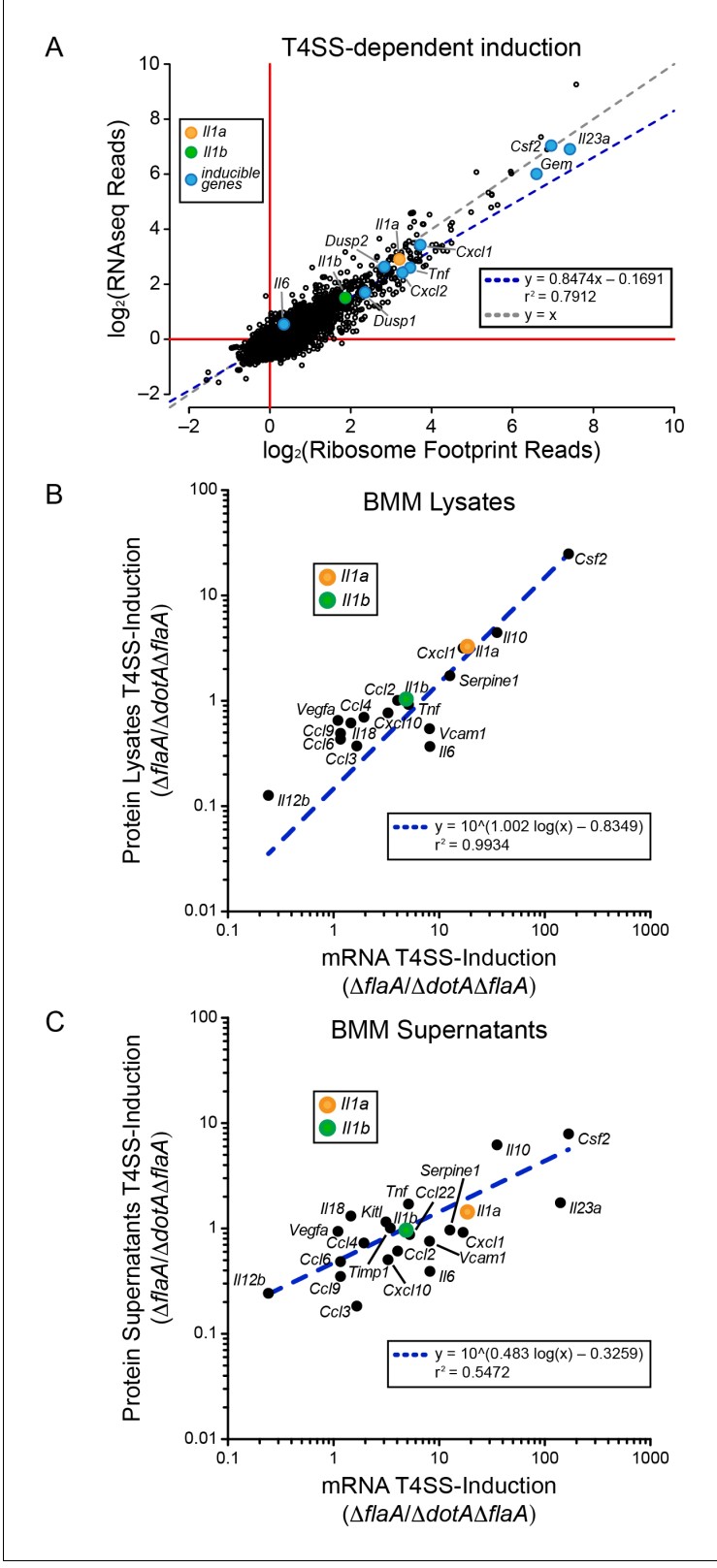

**Figure 2.** mRNA superinduction controls the T4SS-dependent induction of host gene expression in response to *L. pneumophila*. (**A**) The ratio of ribosome footprint and RNAseq read counts for well-expressed transcripts (read count ≥100) in *ΔflaA*-infected versus *ΔdotAΔflaA*-infected B6 BMMs was calculated for each annotated transcript (open circles) in the dataset and plotted. (**B–C**) B6 BMMs were infected with *ΔflaA* or *ΔdotAΔflaA L. pneumophila*

*Figure 2 continued*

and at 6 hr post-infection proteins were measured in cell lysates (B) or supernatants (C) by bead array. The T4SS-induction (*ΔflaA/ΔdotAΔflaA*) of protein in supernatants (B) or lysates (C) and the T4SS-induction of mRNA (*ΔflaA/ΔdotAΔflaA*) was plotted. Proteins were normalized to total protein levels measured by BCA and RNAseq read counts was normalized to transcript length and the sum of their respective mitochondrial protein coding genes. Data are averaged from four (A) or two independent experiments (B–C). Orange circle, *Il1a*. Green circle, *Il1b*. Blue circle, subset of inducible genes. Grey dotted line, y = x. Blue dotted line, linear regression model. r$^2$, coefficient of determination. See also *Figure 2—source data 1*.

The following source data is available for figure 2:

**Source data 1.** Source data from ribosome profiling, RNAseq, and bead array analysis used for *Figure 2*.

seven bacterial effector-dependent induction of inflammatory cytokines occurs by the induction of mRNA transcripts rather than through a mechanism of selective ribosome loading of mRNAs.

## MyD88 signaling in response to *L. pneumophila* is required for mRNA induction

It was previously proposed that specific transcripts, such as *Il1a* and *Il1b*, can be preferentially translated via a mechanism that requires signaling through the adaptor protein MyD88 (*Asrat et al., 2014*). Thus, we performed ribosome profiling and RNAseq on WT and *Myd88*$^{-/-}$ BMMs infected with *ΔflaA L. pneumophila*. For all MyD88-induced genes, including *Il1a* and *Il1b*, we observed a linear correlation between the induction of ribosome footprints and RNAseq reads (*Figure 3B*). This implies that MyD88-dependent induction of *Il1a* and *Il1b* ribosome footprints is controlled primarily at the level of mRNA induction, rather than at the level of selective ribosome loading of the mRNA. A similar pattern was also observed for other MyD88-induced genes, including *Cxcl1*, *Csf2*, *Tnf*, and *Il6* (*Figure 3B*). Taken together, our results argue that the ability of host cells to overcome a pathogen-induced block in protein synthesis, and produce inflammatory cytokines such as IL-1α and IL-1β, requires a T4SS- and MyD88-dependent increase in mRNA levels rather than preferential loading of these cytokine mRNAs with ribosomes.

## Ribosome occupancy of mRNAs varies but is independent of infection

The above analyses sought to determine whether T4SS-dependent or MyD88-dependent gene induction was due to increased mRNA levels or increased ribosome loading of mRNAs. However, the analyses did not reveal whether there is differential ribosome occupancy of constitutively expressed (i.e. non-induced) mRNAs. We thus analyzed the ratio of ribosome footprint reads to RNAseq reads for all (induced and non-induced) transcripts in uninfected BMMs, and in BMMs infected with *ΔflaA*, *ΔdotAΔflaA*, or *Δ7ΔflaA L. pneumophila*. This analysis revealed a wide range of ribosome occupancies across different transcripts (*Figure 4A–D*). As might be anticipated, many of the mRNAs with the highest ribosome occupancy encoded abundant 'housekeeping' proteins, including *Acta1* and histone mRNAs (e.g. *Hist1h2ba*, *H2afj*, and *Hist3h2ba*) (*Figure 4A*; *Table 1*). Importantly, most mRNAs that exhibit increased ribosome occupancy in uninfected BMMs also exhibit increased ribosome occupancy in *ΔflaA*, *ΔdotAΔflaA*, or *Δ7ΔflaA L. pneumophila*-infected BMMs (*Figure 4B–D*; *Table 1*), implying that the increased ribosome occupancy of these mRNAs is constitutive and not induced in response to infection. A few mRNAs of immunological interest, namely *Lyz1*, *S100a11*, and *Cxcl3* exhibited elevated ribosome occupancy in all infection conditions (*Figure 4B–D*; *Table 1*). In contrast, *Ftl1* mRNA exhibited very low ribosome occupancy (*Figure 4A*), consistent with a previous report showing that *Ftl1* translation can be strongly repressed (*Cairo et al., 1989*). *Atf4* is another gene known to be regulated at the level of translation (*Pavitt and Ron, 2012*), and in *ΔflaA* and *ΔdotAΔflaA L. pneumophila*-infected BMMs, *Atf4* exhibited low ribosome occupancy (*Figure 4B–C*). *Atf4* was not expressed at high enough levels to be called as detected in uninfected or *Δ7ΔflaA L. pneumophila*-infected BMMs (*Figure 4A and D*). Taken together, our results reveal that several mRNAs exhibit constitutive increased or decreased ribosome occupancy, as expected. Despite this, ribosome occupancy of mRNAs was not markedly affected by *L. pneumophila* infection (*Figure 4A–D*).

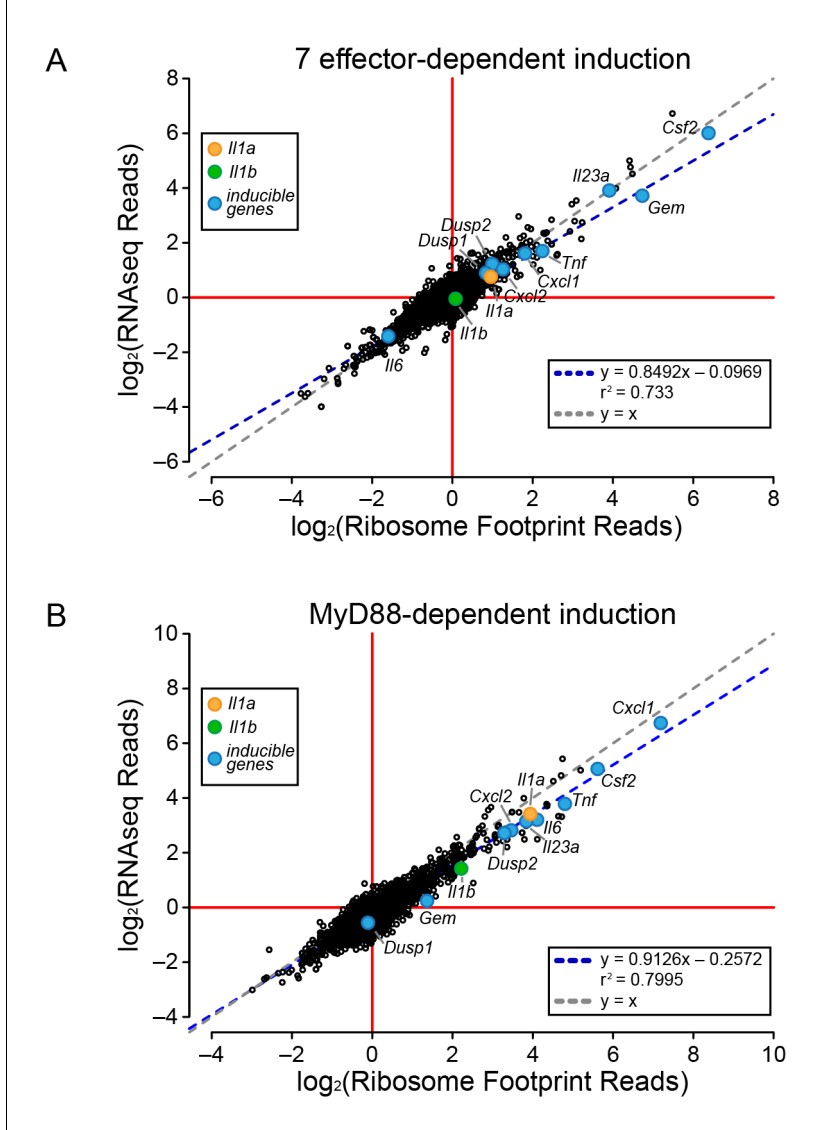

**Figure 3.** Global induction of mRNAs and ribosome footprints in response to *L. pneumophila*. (**A–B**) Ribosome footprint and RNAseq read counts were sorted for well-expressed transcripts (read count ≥100) and normalized to the sum of their respective mitochondrial protein coding genes. The ratio of ribosome footprint and RNAseq read counts in (**A**) *ΔflaA*-infected and *Δ7ΔflaA*-infected B6 BMMs or (**B**) B6 or *Myd88*[−/−] BMMs infected with *ΔflaA L. pneumophila* was calculated for each annotated transcript (open circles) in the dataset and plotted. Data are averaged from two independent experiments. Orange circle, *Il1a*. Green circle, *Il1b*. Blue circle, subset of inducible genes. Grey dotted line, y = x. Blue dotted line, linear regression model. r[2], coefficient of determination. See also *Figure 3—source data 1*.

The following source data is available for figure 3:

**Source data 1.** Source data from ribosome profiling and RNAseq analysis used for *Figure 3*.

## Global analysis of translation inhibition by *L. pneumophila*

A benefit of ribosome profiling is that it permits the mapping of ribosome footprints with nucleotide resolution. Thus, to characterize the position of ribosomes on mRNAs after infection with *L. pneumophila*, we generated metagene ribosome footprint profiles from libraries generated from WT BMMs (*Figure 5*). Metagene profiles were generated by mapping the inferred A site position of ribosome footprint reads relative to the start (*Figure 5A,C,E,G,I*) or stop (*Figure 5B,D,F,H,J*) codon on a

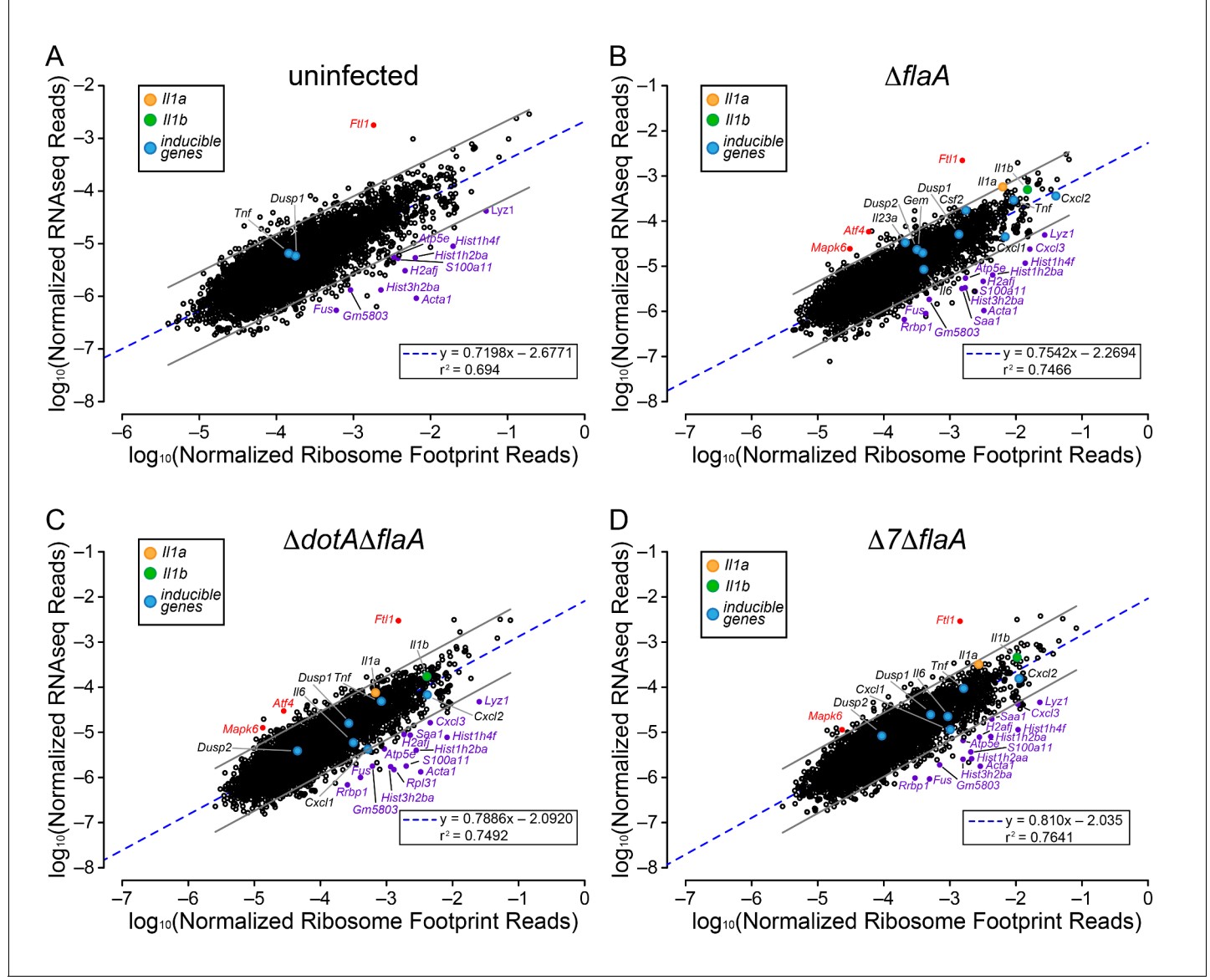

**Figure 4.** Ribosome occupancy does not explain the inducible innate immune response to *L. pneumophila*. (A–D) Ribosome footprint and RNAseq read counts were sorted for well-expressed transcripts (read counts ≥ 100) and normalized to CDS length and the sum of their respective mitochondrial protein coding genes. The normalized read counts for ribosome footprints and RNAseq for all well-expressed annotated transcripts were plotted for (A) uninfected, (B) *ΔflaA*, (C) *ΔdotAΔflaA*, or (D) *Δ7ΔflaA L. pneumophila*-infected B6 BMMs. Red dots represent transcripts with low translation efficiency. Purple dots represent a number of transcripts common to all conditions that appear to have significantly higher ribosome occupancy. Data are averaged from three (A), four (B–C), or two independent experiments (D). Orange circle, *Il1a*. Green circle, *Il1b*. Blue circles, subset of inducible transcripts. Blue dotted line, linear regression model. Grey lines, 99% prediction interval. $r^2$, coefficient of determination. See also *Table 1* and *Figure 4—source data 1*.

The following source data is available for figure 4:

**Source data 1.** Source data from ribosome profiling and RNAseq analysis used for *Figure 4*.

given transcript. Mapped reads were then summed to produce a global view of the distribution of ribosomes across all transcripts in our dataset. As expected from the known stepwise codon-by-codon movement of the ribosome, all metagene plots demonstrated a characteristic three-nucleotide periodicity (*Figure 5*). Interestingly, metagene ribosome profiles of uninfected (*Figure 5A–B*), *ΔflaA* (*Figure 5C–D*), *ΔdotAΔflaA* (*Figure 5E–F*), or *Δ7ΔflaA* (*Figure 5G–H*) infected BMMs

**Table 1.** Transcripts with ribosome occupancy eight times greater than the condition average. Bolded, transcripts found in all conditions. Orange, transcripts found in three conditions. Purple, transcripts found in two conditions. Data are averaged from two independent experiments.

| Uninfected | | ΔflaA | | ΔdotAΔflaA | | Δ7ΔflaA | |
|---|---|---|---|---|---|---|---|
| Gene | Riobosome occupancy | Gene | Riobosome occupancy | Gene | Riobosome occupancy | Gene | Riobosome occupancy |
| **Acta1** | 7105.62 | **Acta1** | 3137.40 | **Acta1** | 2489.70 | **Acta1** | 1633.44 |
| H2-Q7 | 3130.45 | Hist1h4f | 1177.90 | S100a11 | 1115.36 | Hist1h4f | 943.87 |
| **Hist1h4f** | 2195.70 | S100a11 | 870.51 | Hist1h4f | 1069.93 | Rpl31 | 893.93 |
| **Hist3h2ba** | 1715.98 | Hist1h2aa | 844.87 | Rpl31 | 873.61 | Hist1h2aa | 822.96 |
| **H2afj** | 1524.09 | Hist3h2bb-ps | 699.92 | Hist1h2ba | 707.62 | **Hist3h2ba** | 625.90 |
| Hist3h2bb-ps | 1470.88 | **Hist1h2ba** | 692.19 | Hist3h2ba | 670.32 | S100a11 | 565.91 |
| **Lyz1** | 1260.16 | H2afj | 686.47 | **Lyz1** | 533.22 | **Fus** | 532.18 |
| **Hist1h2ba** | 1174.16 | Cxcl3 | 675.23 | Hist3h2bb-ps | 524.22 | **Hist1h2ba** | 519.60 |
| Cd52 | 1170.22 | H2-T24 | 557.88 | **Fus** | 405.23 | **Lyz1** | 498.70 |
| **Fus** | 1102.13 | **Lyz1** | 551.76 | H2-T24 | 374.06 | H2-Q7 | 371.10 |
| H2-Q4 | 1022.17 | Hist3h2ba | 509.95 | Gm5803 | 345.78 | Gm5803 | 368.22 |
| Rpl38 | 1004.74 | **Fus** | 480.25 | H2-Q7 | 337.36 | H2afj | 356.99 |
| Hist2h2ab | 796.60 | Saa1 | 475.25 | Hist1h4i | 302.55 | Hist1h4i | 348.34 |
| H2-Q6 | 752.77 | Gm5803 | 436.48 | Cxcl3 | 281.86 | Hist1h4k | 318.08 |
| **S100a11** | 717.99 | Hist1h4i | 333.72 | Saa1 | 265.08 | Rrbp1 | 306.19 |
| **Gm5803** | 692.32 | Atp5e | 315.00 | Hist1h4n | 244.74 | Hist1h4j | 301.73 |
| Tmsb10 | 679.27 | Rrbp1 | 308.68 | Rrbp1 | 225.85 | Hist1h4a | 298.67 |
| H2-Q10 | 674.79 | Mt1 | 308.36 | Hist1h4j | 218.84 | Hist1h4h | 295.99 |
| Rpl36 | 672.43 | Hist1h4j | 304.87 | Hist1h4k | 217.89 | Hist1h4b | 288.98 |
| Mt1 | 672.29 | Hist1h4k | 303.19 | **Atp5e** | 217.08 | Hist1h4n | 272.04 |
| Hist2h2bb | 650.90 | Hist1h4h | 293.11 | H2-Q4 | 215.94 | BC094916 | 259.08 |
| H2-Q7 | 629.56 | Hist1h4a | 292.51 | Hist1h4h | 215.51 | Hist1h4c | 255.20 |
| H2-Q7 | 618.80 | Hist1h4b | 280.60 | **H2afj** | 206.14 | Cxcl3 | 252.49 |
| **Atp5e** | 606.27 | Gm11127 | 272.59 | Hist1h4a | 205.64 | **Atp5e** | 241.23 |
| H2-T24 | 601.41 | Hist1h4n | 265.41 | Hist1h4b | 197.15 | Saa1 | 220.10 |
| Rpl37 | 584.88 | Fkbp1a | 264.22 | Hist2h2bb | 191.90 | Myl12b | 210.72 |
| H2-T10 | 545.54 | Hist1h4c | Hist1h4c | Hist1h4c | 187.03 | Gm7030 | 206.93 |
| **Hist1h4i** | 529.15 | Gm7030 | 253.47 | Mt1 | 185.28 | | |
| Gm11127 | 512.74 | Myl12b | 247.71 | Cd52 | 184.14 | | |
| Uqcrq | 511.93 | Rps17 | 234.41 | Gm11127 | 183.08 | | |
| Emp1 | 494.39 | Cd52 | 231.77 | Hist1h2bj | 182.74 | | |
| Hist1h2bf | 484.53 | | | Sh3bgrl | 181.81 | | |
| Gm7030 | 481.28 | | | | | | |
| Npc2 | 479.93 | | | | | | |
| Hist1h2bj | 478.33 | | | | | | |
| Usmg5 | 477.21 | | | | | | |
| Hmga2 | 468.10 | | | | | | |

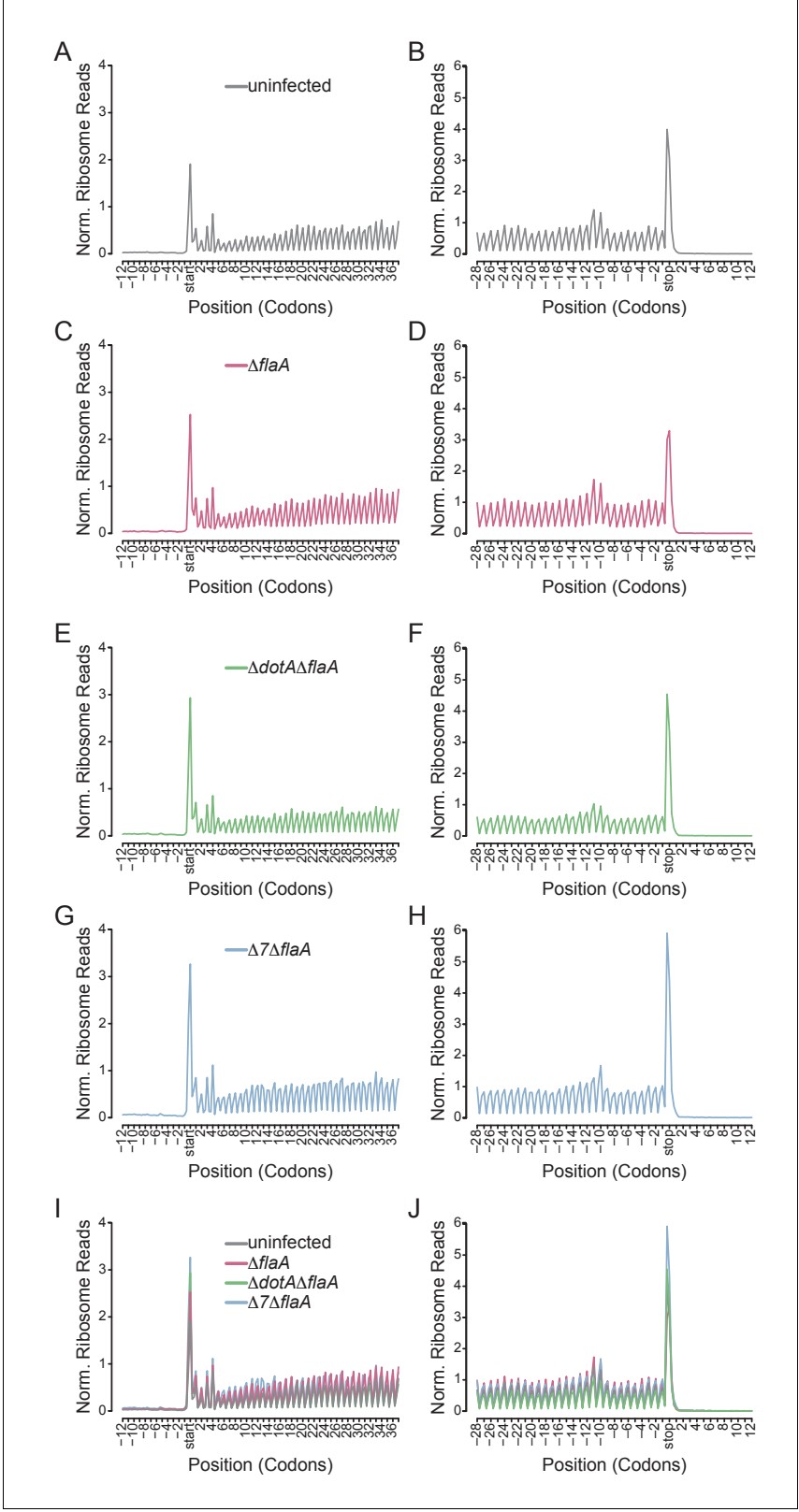

**Figure 5.** Metagene profiles of *L. pneumophila* infected BMMs. (**A–J**) Metagene profiles of uninfected (A–B), *ΔflaA* (C–D), *ΔdotAΔflaA* (E–F), *Δ7ΔflaA* (G–H) *L. pneumophila*-infected B6 BMMs and a merge (I–J). Metagene profiles are depicted relative to the translation start (A, C, E, G, I) and stop site (B, D, F, H, J). Metagene analyses show peaks at every three nucleotides, corresponding to the codon-to-codon shifts of the ribosome. Data are

*Figure 5 continued on next page*

*Figure 5 continued*

representative of two independent experiments (A–J). Black line, uninfected. Red line, *ΔflaA*-infected. Green line, *ΔdotAΔflaA*-infected. Blue line, *Δ7ΔflaA*-infected.

appeared grossly similar to each other, even though global translation is blocked only in the *ΔflaA* and *Δ7ΔflaA L. pneumophila*-infected conditions (*Barry et al., 2013*; *Fontana et al., 2011*; *Asrat et al., 2014*). This result can be explained by the fact that ribosome metagene profiles do not distinguish whether ribosome footprints arise from stalled or translating ribosomes, unless the stall occurs at a characteristic distance from the start or stop codon. In fact, we did notice a slight increase in the number of ribosomes found at the start site of the transcript in *Δ7ΔflaA L. pneumophila*-infected BMMs as compared to other conditions (*Figure 5I*). This may reflect a selective block in translation initiation by this strain (see below). In addition, we noted that in all conditions, ribosomes accumulated at the stop codon, suggesting that, in BMMs, translation termination may be a limiting step in translation (*Figure 5J*).

## *L. pneumophila* blocks translation at the levels of initiation and elongation

To distinguish whether an observed ribosome footprint arises from a stalled or translating ribosome, we performed ribosome run-off experiments. In these experiments, new translation initiation was blocked by the drug harringtonine 120 s prior to cell lysis. Harringtonine inhibits the first rounds of peptide bond formation following ribosome subunit joining and results in accumulation of ribosomes at the translational start site and run-off of elongating (but not stalled) ribosomes that have already cleared the start codon (*Ingolia et al., 2012*, *2011*; *Huang and Harringtonine, 1975*; *Tscherne and Pestka, 1975*; *Fresno et al., 1977*). Importantly, cells experiencing a block in translation elongation will exhibit less ribosome run-off after harringtonine treatment, and an increased number of reads at the 5' end of mRNAs after drug treatment (*Ingolia et al., 2011*), compared to cells in which elongation is not blocked.

As expected, uninfected and *ΔdotAΔflaA*-infected BMMs show an increase in ribosome footprints at the translation start site and a preferential loss of ribosome footprints from the 5' and 3' end of mRNAs, consistent with the expected effects of harringtonine and demonstrating clear ribosome run-off (*Figure 6A–B,E–F*, *Figure 6—figure supplement 1A–B*). By contrast, *ΔflaA L. pneumophila*-infected BMMs treated with harringtonine exhibited little ribosome run-off (*Figure 6C–D*, *Figure 6—figure supplement 1A–B*), consistent with the expectation that *ΔflaA L. pneumophila* blocks host translation elongation. The *Δ7ΔflaA L. pneumophila* strain, lacking all known bacterial effectors that block host protein synthesis, nevertheless, shuts down host translation (*Barry et al., 2013*), yet we observed clear evidence of run-off of elongating ribosomes from the 5' and 3' end of mRNAs following harringtonine treatment (*Figure 6G–H*, *Figure 6—figure supplement 1A–B*). These data suggest that the residual block in host protein synthesis induced by *Δ7ΔflaA L. pneumophila* is at the level of translation initiation. Similar results can be seen when analyzing longer stretches of coding sequences (*Figure 6—figure supplement 1C–F*).

It is important to note that there is a small proportion of uninfected bystander cells assayed in our experiments. However, it is unlikely that these uninfected cells are responsible for the ribosome run-off seen in *Δ7ΔflaA L. pneumophila*-infected BMMs because the conditions used in these experiments led to most (~90%) cells being infected with *L. pneumophila* (*Figure 1—figure supplement 1A–B*). Furthermore, if our infection conditions resulted in large numbers of uninfected cells, then a similar run-off should have been observed in the *ΔflaA*-infected sample, which it was not. Thus, these results suggest that the seven effectors are required to block translation elongation, and that the residual translation inhibition induced by *Δ7ΔflaA L. pneumophila* is at the level of translation initiation (*Figure 6*).

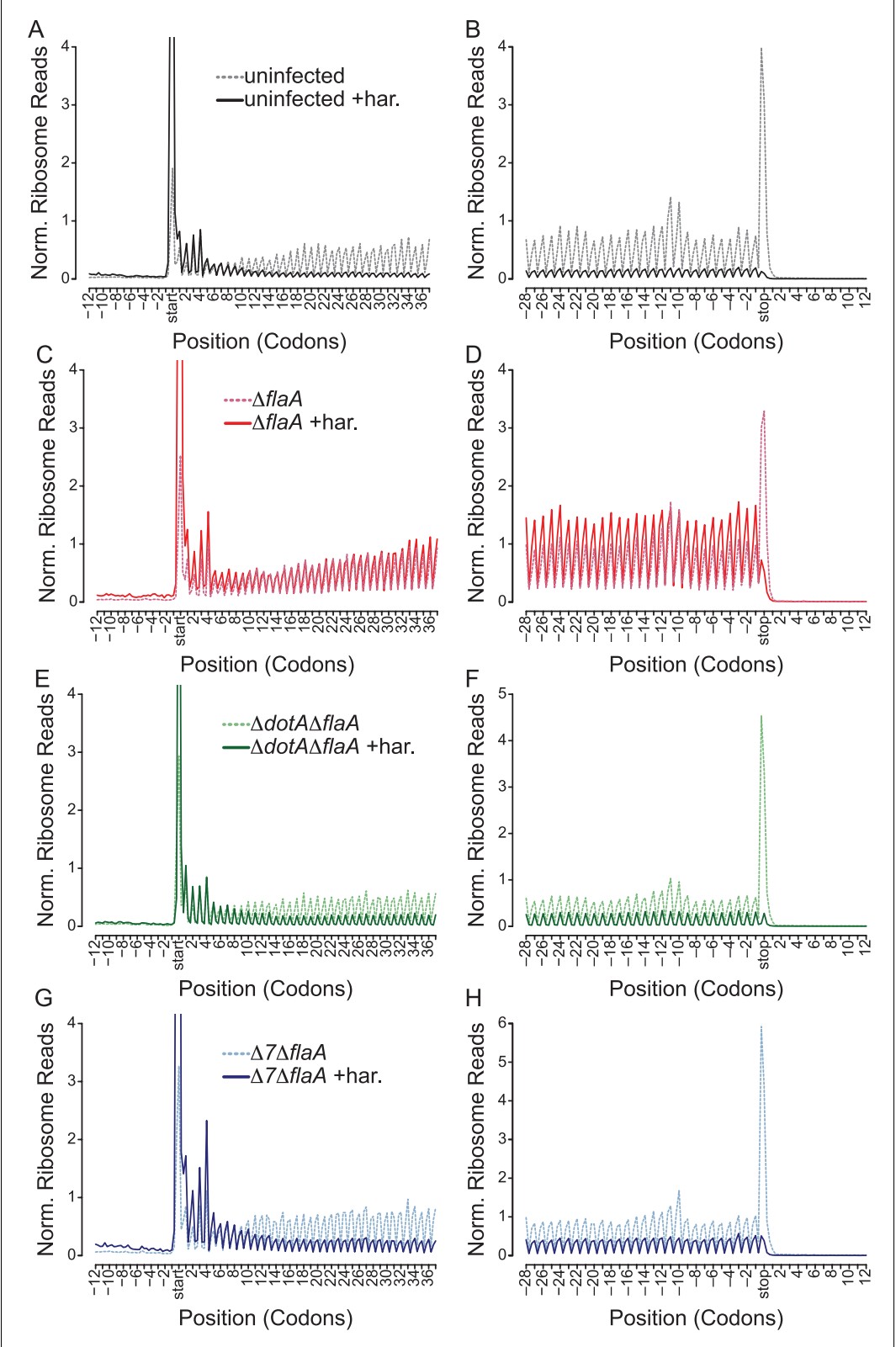

**Figure 6.** *L. pneumophila*-induced block of host protein synthesis occurs at the level of translation initiation and elongation. (**A–H**) Metagene profiles of B6 BMMs uninfected (**A–B**) or infected with *ΔflaA* (**C–D**), *ΔdotAΔflaA* (**E–F**), or *Δ7ΔflaA* (**G–H**) *L. pneumophila* in the presence (solid line) or absence (dashed line) of the drug harringtonine to block translation initiation. Metagene profiles are depicted relative to the translation start (A, C, E, G) and

*Figure 6 continued on next page*

*Figure 6 continued*

stop site (B, D, F, H). Data are representative of two independent experiments (**A–H**). Solid line, no drug treatment. Dashed line, harringtonine treatment. Black line, uninfected. Red line, *ΔflaA*-infected. Green line, *ΔdotAΔflaA*-infected. Blue line, *Δ7ΔflaA*-infected.

The following figure supplement is available for figure 6:

**Figure supplement 1.** *L. pneumophila*-induced block in host protein synthesis can occur at the level of translation elongation and initiation.

## Cytokine transcripts do not escape the pathogen-induced translation block

Although the above results demonstrate a global block in translation elongation in *ΔflaA*-infected cells, it remains possible that specific transcripts escape this block. We therefore analyzed our translation run-off datasets to assess translation elongation on a per-mRNA basis. We plotted the number of ribosome footprint reads for each transcript in paired untreated and harringtonine treated samples (*Figure 7A–D*). In this analysis, we expect that an mRNA with actively elongating ribosomes would show a reduction in the number of 5' reads in the harringtonine treated sample, as ribosomes will run off the transcript, compared to the untreated sample. In order to best measure run-off elongation and avoid the expected but confounding effects of harringtonine-induced accumulation of footprints at start codons (which were clearly observed; *Figure 6*), we excluded the first 25 codons and analyzed ribosome footprint occupancy over the next 300 codons. Consistent with our previous analysis, we find that uninfected, *ΔdotAΔflaA*, and *Δ7ΔflaA*-infected BMMs show a clear global signature of ribosome run-off, again suggesting that the block in host protein synthesis induced by *Δ7ΔflaA L. pneumophila* infection is occurring at the level of translation initiation (*Figure 7A–D*). Importantly, in *ΔflaA*-infected BMMs there is no evidence of ribosome run-off, consistent with *ΔflaA L. pneumophila* inducing a block in host translation elongation (*Figure 7B*). Interestingly, in all conditions tested, cytokine-related genes fell well within the average of ribosome retention across all transcripts, and if anything, were found to have reduced ribosome run-off compared to a typical gene (*Figure 7A–D*). A similar trend was seen when we further examined ribosome run-off for specific immune and housekeeping transcripts by plotting the cumulative read counts over the length of the mRNA (*Figure 7—figure supplement 1*). These results imply that at this time point, cytokine transcripts are not preferentially translated in response to pathogenic infection, but instead are controlled at the level of mRNA induction (*Figure 7A–D*).

## Discussion

Inducible gene expression is of central importance for the immune response to infection. A recent study showed that in response to innate immune stimulation with purified LPS, dendritic cells almost entirely control the induction of genes at the level of transcription (*Jovanovic et al., 2015*). However, this conclusion may not apply to cells infected with a virulent pathogen that manipulates gene expression. We thus investigated the relative contributions of mRNA induction and translation during infection with an intracellular bacterial pathogen, *L. pneumophila*, that blocks host protein synthesis.

Pathogen-induced blockade of host protein synthesis has been shown in a number of infection models to be sensed by the host and induce an inflammatory response (*Barry et al., 2013*; *Fontana et al., 2011*; *Dunbar et al., 2012*; *McEwan et al., 2012*; *Chakrabarti et al., 2012*; *Fontana et al., 2012*). We previously identified IL-1α as a key inflammatory cytokine induced preferentially in response to translation inhibition imposed by *L. pneumophila* (*Barry et al., 2013*). However, the mechanism by which cytokine proteins are induced despite a pathogen-induced translation blockade remains unclear. We previously provided evidence for a model in which translation inhibition results in a failure to synthesize negative feedback inhibitors of transcription, for example, IκB or A20 (*Fontana et al., 2011*). We proposed this results in a massive and sustained production of cytokine transcripts, termed mRNA superinduction, that is sufficient to overcome the partial (~95%) block in translation and allow for production of cytokine proteins (*Barry et al., 2013*; *Fontana et al., 2011*). Another report provided data suggesting that IL-1 production is mediated by MyD88-

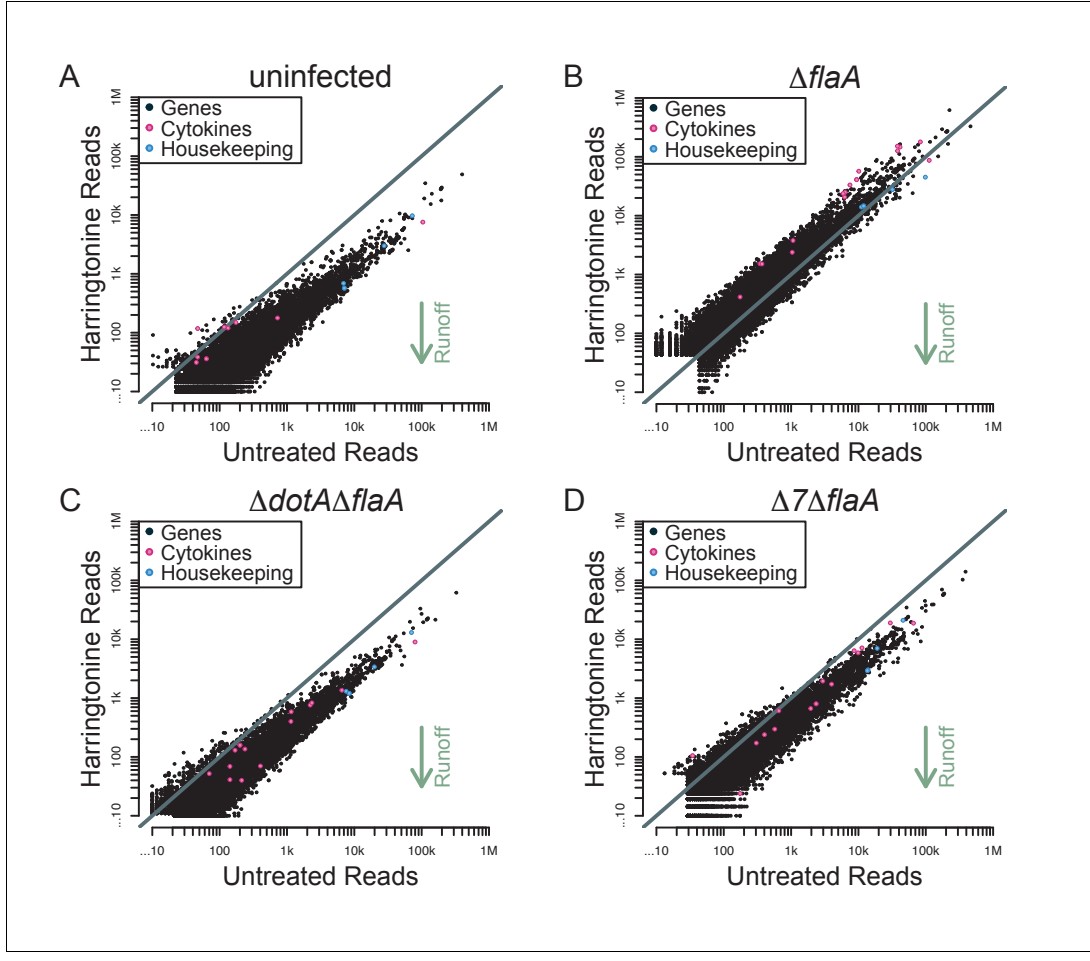

**Figure 7.** Immune-related genes do not have increased translation rates in response to infection with *L. pneumophila*. (A–D) Read counts from paired samples treated with harringtonine or left untreated were plotted for uninfected (A), *ΔflaA* (B), *ΔdotAΔflaA* (C), or *Δ7ΔflaA*-infected (D) BMMs showing where cytokine-related transcripts (pink circles; *Csf1, Csf2, Cxcl1, Cxcl2, Dusp1, Dusp2, Ifnb1, Il10, Il12b, Il1a, Il1b, Il23a, Il6, Lyz1,* and *Tnf*) and housekeeping transcripts (blue circles; *Gapdh, Rpl31, Rps17,* and *Tuba1a*) fall among all transcripts (black circles). Grey line, y=x. Data shown are representative of two independent experiments. See *Figure 7—source data 1* for individual housekeeping and cytokine-related transcripts. Supporting Information Captions.

The following source data and figure supplement are available for figure 7:

**Source data 1.** Source data from ribosome profiling analysis used for *Figure 7*.

**Figure supplement 1.** Individual mRNAs do not show evidence of preferential translation.

---

enhanced protein synthesis, although alternative explanations were also entertained (*Asrat et al., 2014*). A third study proposed that virulent *L. pneumophila* regulates cap-dependent translation initiation, via manipulation of the mTOR signaling pathway, to regulate the protein levels of highly abundant transcripts in infected macrophages (*Ivanov and Roy, 2013*). In our present study, we found that the induction of ribosome footprints by *L. pneumophila* could be explained by an underlying induction of mRNAs. We did not find evidence for selective ribosome loading of abundant cytokine mRNAs. In addition, ribosome run-off experiments confirmed that cytokine mRNAs are not selectively translated during infection (*Figure 7*). Furthermore, we find that the role of MyD88 signaling in gene expression appears to be primarily at the level of mRNA induction and not translational regulation (*Figure 3*). Thus, we conclude that preferential translation does not account for the majority of specific gene induction following infection by *L. pneumophila*.

It remains possible that selective translation initiation mechanisms, for example, via uORFs, might also contribute modestly to the inducible immune response to *L. pneumophila*, but these subtle effects were not evident in our global analysis. In any case, it is difficult to explain how regulation of translation initiation could overcome a downstream pathogen-induced block in translation elongation such as is observed during *L. pneumophila* infection. It is also possible that post-translational mechanisms, which are not addressable with the ribosomal profiling techniques used here, may regulate protein production by infected cells. Indeed, inflammasome-dependent caspase-1 processing is known to be an important post-translational regulatory mechanism controlling IL-1$\beta$ production by infected cells (*von Moltke et al., 2013*). Lastly, our data do not specifically address the mechanism of mRNA induction, although our prior work suggested mRNA induction involves new transcription rather than increased mRNA stability (*Fontana et al., 2011*).

Although wild-type *L. pneumophila* blocks translation elongation via translocated effectors, we found that *Δ7ΔflaA L. pneumophila* lacking the effectors nevertheless blocks protein synthesis at the level of translation initiation (*Figure 6*). Thus, in contrast to a previous study that used virus-based translation reporter experiments in *L. pneumophila*-infected RAW macrophages (*Ivanov and Roy, 2013*), we were clearly able to dissociate the *L. pneumophila*-induced block in host protein synthesis into two components: (1) an elongation block that required the seven translocated effectors, and (2) an initiation block that did not require the seven effectors. In addition, our analysis represents an advance over prior studies because we were able to analyze the translation of all endogenous transcripts simultaneously as opposed to measuring translation only of a single exogenous reporter mRNA. Intriguingly, in contrast to the effector-dependent block in translation that we show occurs at the level of elongation, the majority of host-mediated regulation of translation occurs at the level of translation initiation (*Hershey et al., 2012*). Thus, while it is possible that a novel bacterial effector that directly targets translation initiation could explain the residual inhibition of translation by the *Δ7ΔflaA L. pneumophila* mutant, we favor the hypothesis that the residual block in host protein synthesis may be a result of the host stress response induced by pathogenic infection, consistent with numerous prior studies (*Mohr and Sonenberg, 2012*; *Lemaitre and Girardin, 2013*; *Chakrabarti et al., 2012*; *Ivanov and Roy, 2013*; *Janssens et al., 2014*; *Tattoli et al., 2012*). Indeed, T4SS-competent *L. pneumophila* has been suggested to induce membrane damage that inhibits the mTOR pathway and blocks translation initiation (*Ivanov and Roy, 2013*). Further studies will be required to identify the bacterial and host pathways required for the residual translation inhibition caused by the *Δ7ΔflaA L. pneumophila* mutant.

The results presented here further support a role for translation inhibition as a signal that the innate immune system uses to recognize and preferentially respond to pathogens (*Fontana et al., 2011*). Our work provides nucleotide-level analysis of the global block in host protein synthesis induced by *L. pneumophila*, and demonstrates that *L. pneumophila* infection results in inhibition of host protein synthesis both at the level of translation initiation and elongation. Importantly, our results also provide insights into the molecular mechanisms by which host cells are able to mount a protective immune response despite a pathogen-induced block in protein synthesis. Using ribosome run-off assays in combination with ribosome profiling and RNAseq, we find that mRNA superinduction, rather than selective mRNA translation, is the strategy by which host cells produce inflammatory cytokines in the face of pathogen-mediated translation inhibition. To be effective, the strategy of mRNA superinduction requires that the magnitude of mRNA superinduction exceeds the magnitude of the block in protein synthesis. Indeed, our data suggest this is the case, as we observe >1000 fold induction of certain mRNAs, whereas we previously estimated the block in protein synthesis to be ~95% (20-fold) (*Fontana et al., 2011*). One possible advantage of mRNA superinduction as a strategy for overcoming a pathogen-mediated block of protein synthesis is that it does not require specific translation factors, as was previously proposed might mediate selective mRNA translation in *L. pneumophila*-infected cells (*Asrat et al., 2014*). In addition, in mammalian cells, selective translation is usually regulated at the level of translation initiation, a strategy that would be easily defeated by pathogens such as *L. pneumophila* that block the downstream process of translation elongation. Importantly, since numerous viral and bacterial pathogens and toxins interfere with host protein synthesis, we propose that our results may provide general insight into the inducible innate immune response to infection.

## Materials and methods

### Ethics statement

These studies were carried out in strict accordance with the recommendations in the Guide for the Care and Use of Laboratory Animals of the National Institutes of Health under animal protocol AUP-2014-09-6665. The protocol was approved by the Animal Care and Use Committee at the University of California, Berkeley.

### Access to high-throughput sequencing data

The data discussed in this publication have been deposited in NCBI's Gene Expression Omnibus (Edgar et al., 2002) and are accessible through GEO Series accession number GSE89184 (https://www.ncbi.nlm.nih.gov/geo/query/acc.cgi?acc=GSE89184).

### Cell culture

Macrophages were derived from the bone marrow of C57BL/6J (Jackson Laboratory, Bar Harbor, ME, USA) and $Myd88^{-/-}$ (Hou et al., 2008) mice on the B6 background. Macrophages were derived by 8 days of culture in RPMI supplemented with 10% serum, 100 µM streptomycin, 100 U/mL penicillin, 2 mM L-glutamine and 10% supernatant from 3T3-macrophage-colony-stimulating factor cells, with feeding on day 5. Cells were re-plated in antibiotic free media 24 hr prior to infection with *L. pneumophila*.

### Bacterial strains and infections

All *L. pneumophila* strains were derived from LP02, a streptomycin-resistant thymidine auxotroph derived from *L. pneumophila* LP01. The *ΔdotAΔflaA*, *ΔflaA*, and *Δ7ΔflaA* strains were generated on the LP02 background and have been described previously (Barry et al., 2013; Fontana et al., 2011; Ren et al., 2006; Fontana et al., 2012). Twofold dilutions of *L. pneumophila* strains used for infections were grown overnight in liquid buffered-yeast-extract culture and, at the time of infection, cultures with an optical density (600 nm) greater than 4.0 were selected. BMMs were plated at a density of $1.56 \times 10^5$ cells per cm$^2$ ($1.5 \times 10^6$ cells per well of a six-well plate) and infected at an MOI of 3 by centrifugation for 10 min at 400 x*g*. After 1 hr of infection media was changed. All in vitro *L. pneumophila* infections were performed in the absence of thymidine to prevent bacterial replication which would otherwise differ between the *ΔdotA* and Dot$^+$ strains. The lack of thymidine can result in a loss of bacterial viability, although we attempted to mitigate this concern by examining host gene expression at a relatively early 6 hr time point.

### Library preparation and sequencing for ribosome profiling

Ribosome profiling experiments were undertaken as previously described (Ingolia et al., 2012). BMMs were plated in tissue culture treated six-well plates ($1.5 \times 10^6$ BMMs/well) or 75 cm$^2$ flasks ($1.2 \times 10^7$ BMMs per flask). At 6 hr post-infection BMMs were lysed by flash freezing and thawed in the presence of lysis buffer (Ingolia et al., 2012). When used, harringtonine (LKT Laboratories, Saint Paul, MN) was added at a final concentration of 2 µg/mL for 120 s at the end of the 6 hr infection. 100 µg/mL of cycloheximide (Sigma-Aldrich, St. Louis, MO) was added to freeze ribosomes after the 120 s harringtonine treatment. Following cycloheximide treatment cells were immediately lysed. Clarified lysates were split and some was used to generate ribosome footprints while some was used to isolate total RNA for RNA sequencing (described below). All RNA and DNA gel extractions were performed overnight as previously described (Ingolia et al., 2012). The Ribo-Zero Gold rRNA Removal Kit (Illumina, San Diego, CA) was used to remove rRNA from ribosome profiling samples before the dephosphorylation and linker ligation steps (Ingolia et al., 2012). Final ribosome profiling libraries were sequenced on a HiSeq2000 System (Illumina) with single read 50 (SR50) read lengths by the Vincent J. Coates Genomics Sequencing Laboratory at UC, Berkeley.

### Generation of RNAseq libraries

Clarified lysate was isolated as described above and 300 µL of lysate was mixed with 900 µL of Trizol LS (Thermo Fisher Scientific, Waltham, MA) and RNA was isolated following the manufacturer's guidelines. RNA integrity was measured utilizing the RNA Pico method on the Agilent 2100

Bioanalyzer at the University of California, Berkeley Functional Genomics Laboratory. High-quality RNA with a RNA integrity number (RIN) >8 (Agilent Technologies, Santa Clara, CA) was submitted to the QB3-Berkeley Functional Genomics Laboratory and single read 100 base pair read length (SR100) sequencing libraries were generated. Libraries were sequenced on a HiSeq2000 System (Illumina) by the Vincent J. Coates Genomics Sequencing Laboratory at UC, Berkeley.

## Alignment of RNAseq reads and differential expression analysis

RNA sequencing reads were preprocessed using tools from the FASTX-Toolkit (http://hannonlab.cshl.edu/fastx_toolkit/) by trimming the linker sequence from the 3' end of each read and in some cases removing 10–15 nucleotides from the 5' of each read to mitigate a region of overrepresented nucleotides. Alignment and differential expression analysis of RNAseq reads were undertaken as previously described (*Trapnell et al., 2012*). Briefly, high quality and preprocessed sequencing reads were aligned using the TopHat splicing-aware short-read alignment program to a library of transcripts derived from the UCSC Known Gene data set, and those with no acceptable transcript alignment were then aligned against the *Mus musculus* genome (mm10).

## Alignment of ribosome footprint sequences

Sequences were processed as described previously (*Ingolia et al., 2012*). Sequences were preprocessed by trimming the linker sequence from the 3' end of each sequencing read and removing the first nucleotide from the 5' end of each read. Reads were then aligned to a rRNA reference using the Bowtie short-read alignment program. All sequences aligning the rRNA reference were discarded. All non-rRNA sequencing reads were aligned using the TopHat splicing-aware short-read alignment program to a library of transcripts derived from the UCSC Known Genes data set, and those with no acceptable transcript alignment were then aligned against the mouse genome (mm10). Perfect-match alignments were extracted, and these files were used for analyses. For most analyses, footprint alignments were assigned to specific A site nucleotides by using the position and total length of each alignment, calibrated from footprints at the beginning and the end of CDSes, as previously described (*Ingolia et al., 2012*, *2011*).

## Counting of ribosome profiling and RNAseq reads

Counting of reads was performed as previously described (*Ingolia et al., 2009*, *2011*). Reads were mapped to coding sequences and counted, excluding reads that mapped to the first 15 codons or the last 5 codons of a CDS due to accumulation of ribosomes (*Ingolia et al., 2011*). In order to analyze gene-specific ribosome run-off (*Figure 7A–D*), we counted reads mapping from codon 26 to codon 325, that is, a 300-codon window excluding the first 25 codons of a gene.

## Ribosome occupancy analysis

For analyses of ribosome occupancy (*Figure 4*), ribosome footprint and mRNAseq read counts were calculated similarly. Read counts were normalized to CDS length, as longer transcripts inherently have increased read counts, generating a read density (read density = read count ÷ transcript length) for each gene. Read densities were further normalized to the sum of read counts of 12 mitochondrial protein-coding genes (see below) as an estimate of total cells in each condition, allowing for comparison among different conditions and libraries (*Iwasaki et al., 2016*). For each transcript in the dataset, the average raw ribosome footprint read counts for each infection conditions was calculated and transcripts with an average ribosome footprint or RNAseq read count less than 100 were discarded. Additionally, any transcript that had ribosome footprint reads but 0 RNAseq reads was also discarded. Discarded transcripts were defined as undetectable.

## MyD88-dependent gene induction analysis

Two experiments were used to generate two independent libraries consisting of B6 and *Myd88*⁻/⁻ BMMs infected with *ΔflaA* or *ΔdotAΔflaA L. pneumophila*. For each gene in the dataset, the average raw ribosome footprint read counts for *ΔflaA L. pneumophila*-infected B6 BMMs were sorted and genes with an average ribosome footprint or RNAseq read count less than 100 were discarded. Additionally, any gene that had ribosome footprint reads but no detectable RNAseq reads in B6 or *Myd88*⁻/⁻ BMMs were discarded. The sorted read counts were then normalized to ribosome

footprint or RNAseq read counts of 12 mitochondrial protein-coding genes (see below) as an estimate of total cells in each condition. MyD88-dependent gene induction was calculated using the equation: MyD88-dependent gene induction = average(normalized B6 read count) ÷ average(normalized $Myd88^{-/-}$ read count).

### Type IV secretion system-dependent gene induction analysis

Four independent experiments were used to generate four collections of sequencing libraries consisting of B6 BMMs infected with $\Delta flaA$ or $\Delta dotA\Delta flaA$ L. pneumophila. For each gene in the dataset, the average raw ribosome footprint read counts for $\Delta flaA$ L. pneumophila infected B6 BMMs were sorted and genes with an average ribosome footprint or RNAseq read count less than 100 were discarded. Additionally, any gene that had ribosome footprint reads but no detectable RNAseq reads in $\Delta flaA$ or $\Delta dotA\Delta flaA$ L. pneumophila-infected B6 BMMs were discarded. The sorted read counts were then normalized to ribosome footprint or RNAseq read counts of 12 mitochondrial protein-coding genes (see below) as an estimate of total cells in each condition. T4SS-dependent gene induction was calculated using the equation: T4SS-dependent gene induction = average(normalized $\Delta flaA$-infected read count) ÷ average(normalized $\Delta dotA\Delta flaA$-infected read count).

### Analysis of cytokine and protein levels in infected BMM lysates and supernatants

B6 BMMs were left uninfected or infected with $\Delta flaA$ or $\Delta dotA\Delta flaA$ L. pneumophila at an MOI of 3 in duplicate, as described above. Media was changed 1 hr following infection and at 6 hr post-infection supernatants were collected and BMMs washed with PBS. BMMs were lysed in 400 µL mammalian cell PE lysis buffer (G-Biosciences, St. Louis, MO) following the manufacturers instructions. Lysates and supernatants were cleared by spinning at 20,000 x g for 30 min at 4°C. Cytokine and protein levels were measured using a commercially available cytokine bead array (Rodent MAP 4.0-Mouse Sample Testing, Ampersand Biosciences, Saranac Lake, NY) and total protein levels were measured by bicinchoninic acid (BCA) assay (Ampersand Biosciences, Saranac Lake, NY). Protein and cytokine levels in each infection condition were normalized to total protein levels. Infectivity was confirmed by staining for L. pneumophila (see below). mRNA levels of cytokines were determined by counting (counting method described above) previously acquired RNAseq data of B6 BMMs infected with $\Delta flaA$ or $\Delta dotA\Delta flaA$ L. pneumophila at an MOI of 3 for 6 hr. RNAseq read counts were normalized to transcript length and the sum of RNAseq read counts of 12 mitochondrial protein-coding genes (see below) as an estimate of total cells in each condition (RNAseq normalization described above). T4SS-dependent induction was measured by taking the ratio of protein or mRNA levels in the $\Delta flaA$ infected condition to protein or mRNA levels in the $\Delta dotA\Delta flaA$ infected condition: T4SS-dependent induction = average(normalized $\Delta flaA$ mRNA or protein) divided by average(normalized $\Delta dotA\Delta flaA$ mRNA or protein). T4SS-induction was averaged from two independent experiments and plotted.

### Seven-effector-dependent gene induction analysis

Two independent experiments were used to generate two collections of sequencing libraries consisting of B6 BMMs infected with $\Delta flaA$ or $\Delta 7\Delta flaA$ L. pneumophila. For each gene in the dataset the average raw ribosome footprint read counts for $\Delta flaA$ L. pneumophila-infected B6 BMMs were sorted and genes with an average ribosome footprint or RNAseq read count less than 100 were discarded. Additionally, any gene that had ribosome footprint reads but no detectable RNAseq reads in $\Delta flaA$ or $\Delta 7\Delta flaA$ L. pneumophila-infected B6 BMMs were discarded. The sorted read counts were then normalized to ribosome footprint or RNAseq read counts of 12 mitochondrial protein-coding genes (see below) as an estimate of total cells in each condition. Seven effector-dependent gene induction was calculated using the equation: seven effector-dependent gene induction = average (normalized $\Delta flaA$-infected read count) ÷ average(normalized $\Delta 7\Delta flaA$-infected read count).

### Metagene profile analysis of ribosome profiling libraries

Metagene profiles were generated as previously described (Ingolia et al., 2009, 2011). These metagene profiles indicate the total number of ribosome footprints whose A site falls at the indicated position relative to the start or stop codon of the coding sequence, and reflect a simple, unweighted

sum of the footprint profiles around the beginning and the end of each protein-coding gene. The A site position was estimated for each footprint using a length-dependent offset from the 5' end of the fragment. The distance from this A site position to the start or stop codon of the coding sequence was then computed, taking into account the fact that translation initiation occurs with the second codon in the A site.

## Analysis of ribosome run-off of individual genes

Cumulative ribosome occupancy profiles (*Figure 7—figure supplement 1*) were computed by taking the cumulative sum of ribosome footprints mapping to each position in the gene, scaled by the normalization factor derived from mitochondrial translation in that sample.

## Mitochondrial genes used for library normalization

| Gene ID | Name | Size (bp) |
| --- | --- | --- |
| ENSMUST00000082392 | mt-Nd1 | 299 |
| ENSMUST00000082396 | mt-Nd2 | 326 |
| ENSMUST00000082402 | mt-Co1 | 495 |
| ENSMUST00000082405 | mt-Co2 | 208 |
| ENSMUST00000082407 | mt-Atp8 | 48 |
| ENSMUST00000082408 | mt-Atp6 | 207 |
| ENSMUST00000082409 | mt-Co3 | 241 |
| ENSMUST00000082411 | mt-Nd3 | 96 |
| ENSMUST00000082414 | mt-Nd4 | 439 |
| ENSMUST00000082418 | mt-Nd5 | 588 |
| ENSMUST00000082419 | mt-Nd6 | 153 |
| ENSMUST00000082421 | mt-Cytb | 361 |

## Quantification of infectivity

WT BMMs were plated on a sterile #1.5 coverslip by placing the coverslip in a tissue-culture-treated six-well plates and adding $1.5 \times 10^6$ BMMs/well in antibiotic-free media 24 hr prior to infection. Twofold dilutions of *L. pneumophila* strains used for infections were grown overnight in liquid buffered-yeast-extract culture and, at the time of infection, cultures with an optical density (600 nm) greater than 4.0 were selected. BMMs were infected at an MOI of 3 by centrifugation for 10 min at 400 x*g*. Media was changed after one hour of infection. At 6 hr post-infection coverslips were collected, washed in PBS, and placed in fixative solution (100 uM sodium periodate, 75 uM Lysine, 2.9 uM $NaH_2PO_4$, 3.2% sucrose, and 4% paraformaldehyde) for 1 hr at 37°C. Following fixation BMMs were blocked in 2% goat serum in PBS. To stain extracellular *L. pneumophila*, blocked BMMs were incubated with a rabbit anti-*Legionella* antibody (RRID: AB_231859; Fitzgerald Industries International, North Acton, MA, USA 20-LR45), washed in PBS, and stained with a goat-anti-rabbit IgG secondary antibody conjugated to Cascade Blue (RRID: AB_2536453; ThermoFisher Scientific, Waltham, MA, USA, C-2764). In some experiments, mammalian cell membrane was labeled with FITC-labeled wheat germ agglutinin (Sigma-Aldrich, St. Louis, MO, L4895) prior to permeabilization. BMMs were permeabilized by dipping coverslips into ice-cold methanol. Permeabilized BMMs were blocked with 2% goat serum and stained with a rabbit anti-*Legionella* antibody (Fitzgerald Industries International, North Acton, MA, 20-LR45) followed by incubation with a goat-anti-rabbit IgG secondary antibody conjugated to TexasRed (RRID: AB_2556776; ThermoFisher Scientific, Waltham, MA, T-2767) to mark all (intracellular and extracellular) *L. pneumophila*. Coverslips were mounted in vectashield antifade mounting medium (Vector Laboratories, Burlingame, CA, H-1000) and visualized on a Nikon TE2000 inverted microscope. All antibody stains were incubated for 30 min at 37°C and all blocking steps were incubated for 60 min at 37°C.

Importantly, the staining method described above results in intracellular bacteria staining positive for TexasRed while extracellular bacteria are double positive for Cascade Blue and TexasRed. Quantification of infectivity was undertaken by two methods using the differential staining of intracellular and extracellular *L pneumophila*. In experiments where differential contrast (DIC) microscopy, Cascade Blue, and TexasRed were visualized counting of intracellular bacteria in BMMs was done by hand using the image analysis software ImageJ (RRID:SCR_003070; Rasband, W.S., ImageJ, U. S. National Institutes of Health, Bethesda, Maryland, USA, http://imagej.nih.gov/ij/, 1997–2016) and the Cell Counter plugin (https://imagej.nih.gov/ij/plugins/cell-counter.html). Uninfected BMMs were classified as BMMs that were not associated with *L. pneumophila* or only associated with extracellular (Cascade Blue + TexasRed double positive) bacteria. Infected BMMs were classified as macrophages containing at least one intracellular *L. pneumophila* (Texas Red only), independent of the number of extracellular bacteria associated with the BMM. In experiments where the cell membrane of BMMs was labeled with FITC-conjugated wheat germ agglutinin along with DIC, Cascade Blue, and TexasRed, analysis of infectivity was undertaken using the imaging software Imaris (RRID:SCR_007370; Bitplane, Zurich, Switzerland). Using Imaris, surfaces of BMMs were drawn on the FITC-conjugated wheat germ agglutinin channel to mark individual BMMs. All extracellular bacteria were removed from analysis by generating a new channel that subtracted the Cascade Blue channel from the TexasRed channel, for example Intracellular Channel = TexasRed Channel – (Scaling Value x Cascade Blue Channel). The scaling value was calculated by measuring the average pixel intensities in each channel for double positive bacteria. As an example, if the TexasRed channel had an average pixel intensity of 350 and the Cascade Blue channel was 3500 then the equation would be: Intracellular Channel = TexasRed Channel – (0.1 x Cascade Blue Channel). The outcome of this calculation is the generation of a channel that removes the TexasRed signal of extracellular bacteria, thus allowing for analysis of bacteria that are only intracellular. Lastly, using the Sortomato utility (http://open.bitplane.com/tabid/235/Default.aspx?id=90) in Imaris, new cell surfaces were drawn for cells that contained a signal in the new Intracellular channel (with double positive bacteria removed), marking cells infected with an intracellular bacterium. Surfaces were also drawn for cells that did not have a signal in the Intracellular channel, marking uninfected BMMs or BMMs only associated with extracellular *L. pneumophila*. Results were checked by eye to confirm that all surfaces accurately marked uninfected and infected BMMs; the surfaces generated by Sortomato were used to quantify infectivity.

## Acknowledgements

We thank members of the Vance and Barton Laboratories for discussion and Justin De Leon, Edward Roberts, and Matthew Krummel for technical assistance. We also thank Justin De Leon, Mary Fontana, Shelley Starck, and Jeannette Tenthorey for critical reading and discussion of the manuscript. This work was supported by the HHMI and National Institutes of Health Grants AI063302, AI075039, AI080749, and CA195768. The Vincent J. Coates Genomics Sequencing Laboratory at UC Berkeley is supported by NIH S10 instrumentation Grants S10RR029667 and S10RR027303. REV is an HHMI Investigator, and research in the Vance Laboratory was supported by investigatorships from the Burroughs Wellcome Fund and the Cancer Research Institute.

## Additional information

### Funding

| Funder | Grant reference number | Author |
| --- | --- | --- |
| Cancer Research Institute | | Kevin C Barry<br>Russell E Vance |
| Fibrolamellar Cancer Foundation | | Kevin C Barry |
| National Institutes of Health | CA195768 | Nicholas T Ingolia |
| Howard Hughes Medical Institute | | Russell E Vance |

| | | |
|---|---|---|
| Burroughs Wellcome Fund | | Russell E Vance |
| National Institutes of Health | AI075039 | Russell E Vance |
| National Institutes of Health | AI063302 | Russell E Vance |

The funders had no role in study design, data collection and interpretation, or the decision to submit the work for publication.

### Author contributions

KCB, Conceptualization, Formal analysis, Investigation, Methodology, Writing—original draft, Writing—review and editing; NTI, Conceptualization, Resources, Software, Formal analysis, Investigation, Methodology, Writing—original draft, Writing—review and editing; REV, Conceptualization, Resources, Formal analysis, Supervision, Funding acquisition, Investigation, Methodology, Writing—original draft, Writing—review and editing

### Author ORCIDs

Kevin C Barry, http://orcid.org/0000-0003-1064-5964
Nicholas T Ingolia, http://orcid.org/0000-0002-3395-1545
Russell E Vance, http://orcid.org/0000-0002-6686-3912

### Ethics

Animal experimentation: These studies were carried out in strict accordance with the recommendations in the Guide for the Care and Use of Laboratory Animals of the National Institutes of Health. The protocol was approved by the Animal Care and Use Committee at the University of California, Berkeley.

## Additional files

### Major datasets

The following dataset was generated:

| Author(s) | Year | Dataset title | Dataset URL | Database, license, and accessibility information |
|---|---|---|---|---|
| Barry KC, Ingolia NT, Vance RE | 2017 | Global analysis of gene expression reveals mRNA superinduction is required for the inducible immune response to a bacterial pathogen | https://www.ncbi.nlm.nih.gov/geo/query/acc.cgi?acc=GSE89184 | Publicly available at the NCBI Gene Expression Omnibus (accession no: GSE89184) |

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
