## [Decision Letter]

Thank you for submitting your article "Global analysis of gene expression reveals mRNA superinduction is required for the inducible immune response" for consideration by *eLife*. Your article has been favorably evaluated by Michel Nussenzweig (Senior Editor) and three reviewers, one of whom is a member of our Board of Reviewing Editors. The reviewers have opted to remain anonymous.

The reviewers have discussed the reviews with one another and the Reviewing Editor has drafted this decision to help you prepare a revised submission.

Summary:

Barry, et al., investigate the important issue of how phagocytic cells can raise a cytokine response in the presence of protein synthesis inhibition that occurs in response to pathogens. Previous studies have shown that macrophages hyperinduce cytokine transcripts in response to *Legionella pneumophila* proteins that interfere with host cell protein synthesis inhibitors. In spite of the fact that many of the induced transcripts appear to produce little increase in translation relative to uninfected cells, some of these transcripts are productively translated. Two other labs have forwarded models that the successfully translated products are a consequence of either RNA abundance or perhaps some pattern recognition control of translation. The authors show that the amount of ribosome loading is directly proportional to the transcript abundance, arguing the ability to detect cytokines in the presence of protein synthesis inhibitors is due to massive overexpression of cytokine transcripts. All three reviewers thought the experimental approach was powerful and convincing. However, there was considerable debate as to the novelty of the main conclusion. In particular, Ivanov and Roy, 2013, previously came to the same conclusion that cytokine production in macrophages infected with pathogenic *Legionella* occurred due to high levels of mRNA expression. The present studies clearly strengthen this conclusion substantially, but were not considered to represent a conceptual advance. However, there do appear to be unexpected and interesting aspects to the data that could be explored further, such as the apparent effects of pathogen infection on translational elongation. Following discussion, there was agreement among the reviewers to consider a revised manuscript that further distinguishes this work from the prior studies and addresses experimental weaknesses.

Essential revisions:

1) A more compelling case is required that the findings represent a conceptual advance beyond that of prior work, particularly that of Invanov and Roy. The authors can consider how to best approach this, but one avenue would be to expand investigation/analysis of the control of translational elongation by pathogen infection.

2) The authors should measure protein levels of a set of representative and biologically important host defense cytokines or host defense proteins to establish whether post-translational mechanisms or superinduction is more important for determining actual cytokine/host defense protein levels.

The reviewers considered the issue of whether the elongation block or the initiation block is being overcome by transcript abundance to not be clearly resolved.

One reviewer commented: There is no demonstration in the literature that bypass of the highly expressed proteins is due to bypass of translation elongation. Most of the expression of cytokines is occurring 5-6 hours after infection, and that is also when the translation initiation block is setting in. Also, Roy and coworkers claim that most of the effects are due to effects on translation initiation blockade, so the literature is a little confusing here. It is reasonable to suppose that the expression of cytokines is due to bypass of translation initiation because of the timing, so if they have data that contradicts that point, that is of interest. The second reviewer agreed with this comment.

With respect to the question of more specific direction on revision:

One reviewer responded: In this system, there are two types of translation bypass that can be blocked, either elongation or initiation. With these highly expressed genes, it looks like high level expression is able to overcome the elongation block. If a case can be made for this by looking at a few of these transcripts in detail, that would be a novel contribution. It is even possible they have the data in hand, and that just by analyzing the data in a different fashion they can show this. For instance, by looking at the group of highly expressed transcripts (say the top 8 or so), can they demonstrate a pattern that shows that elongation inhibition (rather than initiation inhibition) is the thing that is being overcome in these transcripts?

The other reviewer responded: The authors have a powerful dataset in hand, and they would be doing themselves a disservice by simply using that data to confirm previous publications in the field. It is likely that within their existing dataset exists additional evidence of translational elongation regulation. Adjusting the text to highlight this finding, and adding some data along the lines suggested by (the other reviewer) should improve this study and assuage my concerns.

---

## [Author Response]

Essential revisions:

1) A more compelling case is required that the findings represent a conceptual advance beyond that of prior work, particularly that of Invanov and Roy. The authors can consider how to best approach this, but one avenue would be to expand investigation/analysis of the control of translational elongation by pathogen infection.

2) The authors should measure protein levels of a set of representative and biologically important host defense cytokines or host defense proteins to establish whether post-translational mechanisms or superinduction is more important for determining actual cytokine/host defense protein levels.

We were gratified that “all three reviewers thought the experimental approach was powerful and convincing”. However, we were unclear on how to “expand investigation/analysis of the control of translation elongation”, and thus sought clarification. The response we received was as follows:

The reviewers considered the issue of whether the elongation block or the initiation block is being overcome by transcript abundance to not be clearly resolved.

One reviewer commented: There is no demonstration in the literature that bypass of the highly expressed proteins is due to bypass of translation elongation. Most of the expression of cytokines is occurring 5-6 hours after infection, and that is also when the translation initiation block is setting in. Also, Roy and coworkers claim that most of the effects are due to effects on translation initiation blockade, so the literature is a little confusing here. It is reasonable to suppose that the expression of cytokines is due to bypass of translation initiation because of the timing, so if they have data that contradicts that point, that is of interest. The second reviewer agreed with this comment.

With respect to the question of more specific direction on revision:

One reviewer responded: In this system, there are two types of translation bypass that can be blocked, either elongation or initiation. With these highly expressed genes, it looks like high level expression is able to overcome the elongation block. If a case can be made for this by looking at a few of these transcripts in detail, that would be a novel contribution. It is even possible they have the data in hand, and that just by analyzing the data in a different fashion they can show this. For instance, by looking at the group of highly expressed transcripts (say the top 8 or so), can they demonstrate a pattern that shows that elongation inhibition (rather than initiation inhibition) is the thing that is being overcome in these transcripts?

The other reviewer responded: The authors have a powerful dataset in hand, and they would be doing themselves a disservice by simply using that data to confirm previous publications in the field. It is likely that within their existing dataset exists additional evidence of translational elongation regulation. Adjusting the text to highlight this finding, and adding some data along the lines suggested by (the other reviewer) should improve this study and assuage my concerns.

We feel there still may be some confusion, but we will try to address each of the major points above as clearly as possible in turn:

Essential Revision 1:We are asked to better clarify the conceptual advance of our work over that of Ivanov and Roy. In particular, the reviewers state “There was considerable debate as to the novelty of the main conclusion. In particular, Ivanov and Roy, 2013, previously came to the same conclusion that cytokine production in macrophages infected with pathogenic *Legionella* occurred due to high levels of mRNA expression. The present studies clearly strengthen this conclusion substantially, but were not considered to represent a conceptual advance.”

Overall Response:The fundamental unresolved question at issue is: how do macrophages infected with *Legionella* make cytokines despite experiencing a pathogen-induced block in protein synthesis? There appears to be some confusion over what the Ivanov and Roy paper is claiming. In our opinion, Ivanov and Roy invoke a very different model – involving selective or preferential translation of specific mRNAs – than the model we propose in our manuscript, which does not involve preferential translation. One reason for the confusion may be that the Ivanov and Roy paper is primarily focused on modulation of mTOR by *Legionella* and thus is not entirely clear on the (side) question of how cytokine mRNAs are being translated. Most importantly, however, we believe that our paper is the only one that provides the experimental data required to distinguish various possible models to explain cytokine induction.Thus, we feel our paper represents a substantial advance over Ivanov and Roy.

Detailed Response:

In their Abstract, Ivanov and Roy state: “Detection of pathogen signatures resulted in translational biasing toward proinflammatory cytokines through mTOR-mediated regulation of cap-dependent translation.” In our opinion, the invocation of “translational biasing” and specific effects on cap-dependent translation implies a model in which there is selective or preferential translation of specific cytokine transcripts. This is in polar opposition to our model, which proposes there is no preferential translation or translational biasing: indeed, our data demonstrate that all mRNAs are translated indistinguishably. Thus, we propose a model in which mRNA abundance, not translation efficiency, explains differential protein production.

There are several other sections of the Ivanov and Roy paper that make it clear that they are favoring a model in which there is preferential translation of pro-inflammatory cytokines (e.g., IL-6) versus anti-inflammatory cytokines (e.g., IL-10). For example, consider this passage from their paper:

“Although there were substantial differences in cytokine production in response to virulent *L. pneumophila* and the avirulent *L. pneumophila dotA* mutant, these differences did not correlate with gene expression. The amount of *Il6* mRNA was similar under all infection conditions, and *Il10* mRNA was more abundant after infection with virulent *L. pneumophila* even though IL-10 production was lower (Figure 2). Thus, the cytokine biasing observed in response to virulent *L. pneumophila* involved post-transcriptional regulation of gene expression.”

Later, on p. 1224, Ivanov and Roy even propose a specific translation mechanism to explain the differential translation of IL-6 vs IL-10, stating that “IL-10 production was more sensitive to inhibition of eIF4E than was IL-6 production…Thus, cytokine biasing in LPS-treated macrophages was achieved by modulation of the cap-dependent translation initiated by eIF4E.” We note that Ivanov and Roy propose that differential cap-dependent translation regulates IL-6 vs. IL-10 production without actually examining the translation rates of IL-6 vs. IL-10. Nevertheless, in our opinion, these excerpts clearly illustrate that Ivanov and Roy are proposing a preferential translation model, which is entirely distinct from the model we propose. We therefore do not agree with the reviewers’ comment that “Ivanov and Roy, 2013, previously came to the same conclusion that cytokine production in macrophages infected with pathogenic *Legionella* occurred due to high levels of mRNA expression” and we do not agree that we are “simply using [our] data to confirm previous publications in the field.”

Despite the above potential disagreement about what Ivanov and Roy are claiming, we feel the most important point is that the only way to reach a conclusion about whether or not there is translational biasing is by a direct examination of the rate at which cytokine mRNAs are translated in infected cells.It is very important to emphasize that Ivanov and Roy perform no experiments whatsoever to address the translation rate of cytokine mRNAs. Thus, although they claim in their Abstract and elsewhere that they favor a model in which there is translational biasing, there is no experimental support for such a model in their paper. By contrast, we perform detailed ribosome profiling of infected cells, including experiments in which we examine ribosome run-off in harringtonine-pulsed cells. These data allow us for the first time to examine the translation rate across individual cytokine mRNAs in infected cells and allow us (and us alone) to conclude there is no evidence for preferential translation of cytokine mRNAs in infected cells.

A second related issue raised by reviewers was whether our data might refute the claim of Roy and Ivanov that the primary translational block in infected cells is a block of translation initiation. The reviewers suggest that “by looking at the group of highly expressed transcripts (say the top 8 or so), can they demonstrate a pattern that shows that elongation inhibition (rather than initiation inhibition) is the thing that is being overcome in these transcripts.”

We agree that Ivanov and Roy believe the primary translation block in infected cells is at the level of translation *initiation*. By contrast, our data demonstrate that cells infected with wild-type *Legionella* experience a block in *both* translation initiation *and* elongation. Thus, this is another important distinction between our paper and the Ivanov and Roy paper and further substantiates our claim (above) that the two papers are presenting very different models/conclusions.

In fact, we believe that one of the significant aspects of our paper is that we are able to experimentally and genetically dissect the two distinct mechanisms *Legionella* uses to block translation. By performing ribosome run-off experiments we observe that in wild-type infected cells, ribosomes do not run off of (i.e., elongate across) mRNAs, whereas we do see robust run-off (elongation) in uninfected cells or cells infected with the avirulent ∆*dotA*∆*flaA* mutant. Importantly, the block in elongation we observe depends entirely on effectors translocated by *Legionella*, as a strain lacking these effectors (∆*7*∆*flaA*) did not block translation elongation. However, since this strain still blocks translation (as previously reported by us and several other groups, including Ivanov and Roy), we conclude that the block by the ∆7 strain is at the level of translation initiation (and probably involves inhibition of cap-dependent translation mTORC1, as proposed by Ivanov and Roy). Our model is thus that *Legionella* blocks elongation via translated effectors, and blocks initiation via a 7-effector-independent mechanism.Importantly, although Ivanov and Roy provide evidence for an effector-independent block of translation *initiation*, they do not address the effector-dependent block in *elongation*, and thus, they do not experimentally or genetically distinguish the two levels of translational blockade that are imposed by *Legionella* (as we do). This is an important point because it implies that the ability of cells to make cytokine is robust to pathogen-induced blockade of translation, regardless of whether it is elongation, or initiation, or both, that is blocked. In essence our data provide an important mechanism by which hosts could potentially circumvent any mechanism pathogens use to disrupt host protein synthesis.

If we have interpreted the reviewers’ comments correctly, it seems the reviewers appreciate the above distinction between our paper and that of Ivanov and Roy, and appreciate that it is an important distinction. The reviewers also seem to appreciate that an important distinction between our paper and the Ivanov and Roy paper is that we examined translation rates of individual mRNAs, whereas the analysis of translation in the Ivanov and Roy paper is simply bulk analysis (e.g., puromycin incorporation). We believe it is appreciation of this latter point that leads the reviewers to ask us to demonstrate that highly expressed cytokine mRNAs are experiencing a block in elongation in addition to the block in initiation. However, we believe these data are already provided in the manuscript in Figure 7 and Figure 7—figure supplement 1 and 2, where we quantify the number of ribosome footprints in the presence and absence of harringtonine. Ongoing translation elongation permits ribosome run-off during harringtonine treatment and this is reflected in a decrease in ribosome footprints after harringtonine treatment in the uninfected, ∆dot and ∆7 conditions. Importantly, such runoff is not observed in WT-infected cells, implying that there is a block in translation elongation in these cells. Importantly, the lack of run-off (i.e., translation elongation) is apparent for highly expressed transcripts as well as moderately or low expressed transcripts (as shown in Figure 7, where all genes lie on the diagonal).

In summary: our data demonstrate that *Legionella* blocks translation *initiation* and *elongation* and that this dual block is apparent among highly expressed transcripts as well as more moderately expressed transcripts. Thus, we conclude that the ability of *Legionella*-infected cells to produce cytokine proteins is not related to selective escape of the block in translation elongation or initiation, but is instead due to the dramatic upregulation of cytokine mRNA levels.

In relation to the issue of elongation vs. initiation, the reviewers made a few other related comments that we would like to respond to with the hope that it will help clarify our position:

“There is no demonstration in the literature that bypass of the highly expressed proteins is due to bypass of translation elongation.”

We somewhat agree with this point. Certainly, as discussed above, the Ivanov paper does not address whether *Legionella* inhibits translation elongation. However, there are other studies showing that *Legionella* effectors target host elongation factors (e.g., modification of eEF1 via direct glucosylation). Given these past results, we think it is reasonably well accepted that translation elongation is blocked in *Legionella*-infected cells, and thus, the ability of infected cells to produce proteins must require some kind of ‘bypass’ of an elongation block. However, we agree with the reviewer that our paper is the first to demonstrate that there is both an effector-induced block in elongation and a 7-effector-independent block in initiation.

As an aside, we would also like to add that the term ‘bypass’ might be confusing and think it should be avoided. We do not literally think that host cells encode a mechanism to somehow reverse a block in elongation, i.e., remove the glucose residue from the elongation factor, or somehow allow ribosomes to function without an elongation factor. Instead, we believe that the way cells ‘bypass’ the elongation block (which is a ~95% block, not a 100% block) is by superinducing cytokine mRNAs. Most of these mRNAs are also likely blocked, but since the cytokines are induced in some cases by 3 logs, this is sufficient to overcome the ~1 log block in translation.

“Most of the expression of cytokines is occurring 5-6 hours after infection, and that is also when the translation initiation block is setting in.”

We agree with this. However, it is also important to note that at this time point, the block in elongation has already occurred. So, both elongation and inhibition are blocked, and thus, if proteins are made at this time point, then both blocks must be overcome.

“Also, Roy and coworkers claim that most of the effects are due to effects on translation initiation blockade, so the literature is a little confusing here.”

Yes, we agree that the literature in general (and the Roy paper in particular) is confusing. Roy and coworkers do suggest there is a block in initiation, but they do not rule out or address whether there is also a block in elongation. Indeed, their methodology does not allow them to distinguish whether there is a transcript-specific block in initiation or elongation (and, indeed, as discussed above, our ability to do so is one of the important issues our work clarifies).

“It is reasonable to suppose that the expression of cytokines is due to bypass of translation initiation because of the timing, so if they have data that contradicts that point, that is of interest.”

Yes, we believe that our data clearly show that at 6h post infection the cells are experiencing both a block in elongation and initiation, and we do agree with the reviewer that this is a point of interest.

Essential Revision 2:The reviewers ask us to examine protein levels of cytokines to address whether mRNA induction or post-translational mechanisms contribute most to the ultimate levels of cytokine proteins.

To address the request of the reviewers, we infected macrophages with wild-type or ∆*dot* mutant *Legionella* for 6hr and then measured levels of 42 cytokines or other proteins in the supernatant or in cell lysates using commercially available bead arrays (Rodent MAP 4.0-Mouse Sample Testing, Ampersand Biosciences, Saranac Lake, NY, USA). Of the cytokines assayed, 18 cytokines/proteins were measured above the limit of detection in lysates, and 22 cytokines/proteins were above the limit of detection in supernatants. We plotted the T4SS-dependent fold-induction of these cytokine protein levels against the T4SS-dependent fold-induction of the cytokine mRNA levels. The results are now shown in new Figure 2. The results show a remarkably robust correlation between the extent of mRNA induction and the extent of protein induction, particularly in lysates. The correlation seems to apply for the most highly induced proteins/mRNAs (e.g., IL-10 and GMCSF (=Csf2)) but also for more modestly induced cytokines (Il1a, Il1b, Cxcl10, etc.). The correlation is somewhat less robust when looking at protein levels in the cell supernatant, but this may reflect differing rates of secretion, accumulation in the supernatant over time, re-binding to cell surface receptors, and stability in the supernatant. Nevertheless, taken together, the results are consistent with our overall conclusion that the inducible immune response to *Legionella* is due primarily to mRNA superinduction.